# Engineered Human Dental Pulp Stem Cells with Promising Potential for Regenerative Medicine

**DOI:** 10.3390/biotech14040088

**Published:** 2025-11-03

**Authors:** Emi Inada, Issei Saitoh, Masahiko Terajima, Yuki Kiyokawa, Naoko Kubota, Haruyoshi Yamaza, Kazunori Morohoshi, Shingo Nakamura, Masahiro Sato

**Affiliations:** 1Department of Pediatric Dentistry, Graduate School of Medical and Dental Sciences, Kagoshima University, Kagoshima 890-8544, Japan; inada@dent.kagoshima-u.ac.jp (E.I.); kubonao@dent.kagoshima-u.ac.jp (N.K.); hyamaza@dent.kagoshima-u.ac.jp (H.Y.); 2Department of Pediatric Dentistry, Asahi University School of Dentistry, Gifu 501-0296, Japan; isaitoh@dent.asahi-u.ac.jp (I.S.); ykiyokawa@dent.asahi-u.ac.jp (Y.K.); 3Department of Anatomy, Asahi University School of Dentistry, Gifu 501-0296, Japan; terajima5708@gmail.com; 4Division of Biomedical Engineering, National Defense Medical College Research Institute, Saitama 359-8513, Japan; kmorohoshi@ndmc.ac.jp (K.M.); snaka@ndmc.ac.jp (S.N.); 5Department of Genome Medicine, National Center for Child Health and Development, Tokyo 157-8535, Japan

**Keywords:** multi-differentiation, transdifferentiation, dental pulp stem cells (DPSCs), stem cells, gene-engineered, transfection, iPSCs, scaffold, tissue engineering, organoid

## Abstract

The fields of regenerative medicine and stem cell-based tissue engineering hold great potential for treating a wide range of tissue and organ defects. Stem cells are ideal candidates for regenerative medicine because they are undifferentiated cells with the capacity for self-renewal, rapid proliferation, multilineage differentiation, and expression of pluripotency-associated genes. Human dental pulp stem cells (DPSCs) consist of various cell types (including stem cells) and possess multilineage differentiation potential. Owing to their easy isolation and rapid proliferation, DPSCs and their derivatives are considered promising candidates for repairing injured tissues. Recent advances in gene engineering have enabled cells to express specific genes of interest, leading to the secretion of medically important proteins or the alteration of cell behavior. For example, transient expression of Yamanaka’s factors in DPSCs can induce transdifferentiation into induced pluripotent stem cells (iPSCs). These gene-engineered cells represent valuable candidates for regenerative medicine, including stem cell therapies and tissue engineering. However, challenges remain in their development and application, particularly regarding safety, efficacy, and scalability. This review summarizes current knowledge on gene-engineered DPSCs and their derivatives and explores possible clinical applications, with a special focus on oral regeneration.

## 1. Introduction

Stem cells are defined as cells with the capacity to differentiate into various types of cells, a property known as multilineage differentiation or multi-differentiation, along with active proliferative activity (self-renewal) [1]. Several types of stem cells exist, including embryonic stem cells (ESCs), adult somatic stem cells (mesenchymal, hematopoietic, neuronal, and endothelial stem cells), and germ cell-specific stem cells, such as spermatogonial cells in the testis [2,3,4]. Induced pluripotent stem cells (iPSCs) and induced tissue-specific stem cells (i-TSCs), both generated by transfecting adult differentiated somatic cells with Yamanaka’s factors, are also valuable resources for regenerative medicine [2,3,4].

Bone marrow stem cells (BMSCs), derived from the bone marrow, are amongst the most widely used stem cells in clinical settings. They exhibit continuous proliferative capacity and multipotency, with the ability to differentiate into various types of cells, including osteoblasts, chondrocytes, adipocytes, tenocytes, and cardiomyocytes [5]. However, BMSCs are associated with high cost and an invasive collection procedure [6]. Dental pulp stem cells (DPSCs) are a population of cells, called dental pulp cells (DPCs), the resident mesenchymal stromal cells within the pulp cavity of impacted third molars [7] (Figure 1). Human DPSCs are ectodermal-derived stem cells originating during tooth development from ectodermal cells that migrate from the neural tube to the oral region [8,9,10]. Since their first isolation and characterization by Gronthos et al. [11], multiple reports have demonstrated the potential of DPSCs to differentiate into diverse cell types, including osteoblasts, adipocytes, chondrocytes, myocytes, neuronal and endothelial cells, hepatocytes, and melanocytes [12,13].

As mentioned above, DPSCs have potential as an alternative to BMSCs for stem cell-based therapies. However, whether DPSCs are comparable to BMSCs at the molecular level remains unclear. Table 1 presents a comparison of the characteristics of DPSCs and BMSCs based on the studies of Gimble et al. [14] and Ponnaiyan et al. [15]. Both DPSCs and BMSCs are multipotent cells that share many characteristics. However, key differences exist in their specific surface marker expression, differentiation capabilities, and proliferation rates. For example, both cell types strongly express the standard MSC-associated markers CD29 (integrinβ1 chain), CD44 (non-kinase transmembrane glycoprotein), CD73 (5′ectonucleotidase), CD90 (Thy1), and CD105 (endoglin). They typically lack expression of the hematopoietic and endothelial markers CD11b (integrin alpha-M), CD14 (lipopolysaccharide-binding protein), CD34 (transmembrane phosphoglycoprotein protein), and CD45 (lymphocyte common antigen), as well as the major histocompatibility complex class II (HLA-DR) marker. Interestingly, Kim et al. [16] used RNA sequencing (RNA-seq) to compare gene expression profiles in DPSCs and BMSCs and found distinct patterns of gene expression between the two cell types. Kumar et al. [17] demonstrated that DPSCs are a superior stem cell source for obtaining neural progenitor cells. DPSCs also exhibit greater proliferative potential than BMSCs [18,19]. Furthermore, DPSCs display stronger odontogenic capability than BMSCs, suggesting that they may be more suitable for tooth regeneration [20]. Compared with other adult stem cells, such as BMSCs, DPSCs are superior due to their easy, ethical, and non-invasive collection from extracted teeth, along with their high proliferative and multilineage differentiation capabilities, which make them a unique choice for regenerative medicine.

Engineered cells are genetically modified cells generated by adding, deleting, or altering genetic sequences at specific target sites in living cells [21]. As a result, these engineered cells may exhibit inactivation of target genes, acquisition of novel genetic traits, and correction of pathogenic gene mutations. They can also be used to explore the function of a gene of interest (GOI), introduce new characteristics (such as enhancement of proliferation or differentiation ability), secrete therapeutic proteins or small particles useful for regenerative medicine, or induce phenotypic changes, such as transdifferentiation and reprogramming into iPSCs or iTSCs [2,3,4]. These engineered cells, therefore, represent an attractive option for stem cell-based therapies.

In this review, we summarize the current knowledge on DPSCs with a special focus on the genetic engineering of primarily isolated dental cells and their use in studies related to regenerative medicine. We also discuss potential clinical applications of engineered DPSCs and their derivatives, particularly emphasizing the generation of three-dimensional (3D) bioengineered teeth embedded in various scaffold biomaterials and the subsequent grafting of these structures for regenerating damaged dental tissues.

## 2. Isolation and Culture of DPSCs

To date, various types of stem cells have been identified and isolated from oral tissues in the dental field, which serve as promising sources for regenerative therapies. In addition to DPSCs, which are present in the dental pulp, several other stem cells populations have been reported, including stem cells from human exfoliated deciduous teeth (SHED) [18], periodontal ligament (periodontal ligament stem cells, PDLSCs) [22], dental follicle (dental follicle stem cells, DFSCs) [23], apical papilla, (stem cells apical papilla, SCAP) [24,25], human deciduous teeth-derived DPCs (HDDPC) [26], and gingival stem cells (GING SCs) [27].

### 2.1. DPSCs from Enzymatic Digestion of DP Tissue (DPSC-ED) vs. DPSCs Obtained Through Outgrowth from Tissue Explants (DPSC-OG)

DPSCs can be isolated from either deciduous or permanent teeth. After tooth extraction, the tooth surface is wiped with 70% ethanol and rinsed with sterile water. The tooth is then cut to expose the pulp cavity, and dental pulp tissue is obtained by osteotomy (using osteotomy forceps or osteotomy pliers). There are two main approaches for DPSC dissociation: direct isolation from pulp tissue (DPSC-ED) and the outgrowth method using tissue explants (DPSC-OG) [28,29]. The former is a common method involving mechanical extraction of the soft pulp connective tissue, followed by maceration and enzymatic digestion with collagenase and dispase to release the cells from the dental pulp tissue. The isolated cells are then seeded onto plastic culture plates. Although the number of cells obtained tends to be low, this method allows the isolation of relatively homogenous cell populations. In contrast, the explant method allows cells to migrate out of the pulp tissue [30]. Hilkens et al. [30] reported that pulp tissues isolated are directly placed on culture plates, and cells gradually migrate outward. This approach generally requires more time than DPSC-ED. Comparisons between the two methods indicate that DPSC-ED tends to exhibit higher mineralization capacity, while both methods yield cells capable of adipogenic, chondrogenic, and osteogenic differentiation [30]. In DPSC-OG, although the number of recovered cells is higher, this method may select for specific subpopulations, thereby decreasing population homogeneity. Recently, Kiyokawa et al. [31] established a novel approach called scratch-based isolation of primary cells from human dental pulps (SCIP), which is conceptually similar to DPSC-OG. In this method, dental pulp tissue, briefly digested with proteolytic enzymes, is scratched onto a culture dish and grown in a suitable medium. By day 2, cells begin to spread radially from the pulp, and by day 10, they reach a semi-confluent state. Cells harvested by trypsinization consistently yield over 1 million cells, and upon re-cultivation, they can be propagated for more than ten passages. Importantly, their proliferative and differentiation capacities after the 10th passage remain comparable to those of early-passage cells. These isolated cells continue to express stem cell-specific and differentiation-related markers, even after extended passage. The simplicity, high success rate, and adaptability of SCIP, including its applicability to patients with genetic diseases, make it a valuable method for regenerative medicine research and clinical applications.

### 2.2. Optimizing Conditions for DPSCs in Cell Culture

Current techniques for expanding DPSCs require the use of fetal bovine serum (FBS). In particular, medium containing 20% FBS is often considered optimal for DPSC cultivation [29,32]. However, animal-derived reagents (such as FBS) pose safety concerns for clinical therapy, including risks of infection and immunological reactions following transplantation. In addition, the proliferation and differentiation of DPSCs vary considerably, depending on the serum batches. Expanding DPSCs in medium supplemented with human serum (HS) may overcome these problems because HS has been shown to support better proliferation of DPSCs than FBS [33]. Furthermore, inclusion of 1% HS in a chemically defined medium was also beneficial for DPSC proliferation. The type of serum also affects the differentiation potential of the cells. Khanna-Jain et al. [33] demonstrated that cultivation of DPSCs in HS-containing medium promoted more pronounced adipogenic and osteogenic differentiation, while FBS-containing medium supported more effective chondrogenic differentiation. Based on these findings, HS appears to be a suitable alternative to FBS for the safe expansion of DPSCs, although there is a possibility that serum batches can significantly affect experimental results.

### 2.3. Requirement for Xeno-Free Cell Culture System of DPSCs for Human Cell Therapy

Animal-derived components have traditionally been widely used in stem cell cultures. However, serious concerns remain regarding using the cultured DPSCs for human cell therapy, such as the risk of viral, bacterial, fungal, and prion contamination as well as the possible induction of immune rejection of the transplanted cells into the host [34]. For these reasons, the development of xeno-free cell culture protocols that meet Good Manufacturing Practice (GMP) standards has been continuously tested for stem cell culture [34].

Fujii et al. [7] succeeded in cultivating DPSCs using serum-free medium, STK2 [containing Iscove’s modified Dulbecco’s medium (IMDM), recombinant human albumin, hydrocortisone, and an optimal mix of growth factors, including basic fibroblast growth factor (bFGF)], a novel serum-free medium developed for human MSC multiplication. When compared with DPSCs in control medium [Dulbecco’s modified Eagle’s medium (DMEM) containing 10% FBS], those cultured in STK2 exhibited active proliferation. Furthermore, DPSCs grown in STK2 induced to undergo osteogenesis exhibited alkaline phosphatase (ALP) activity and calcification, indicating maintenance of differentiation potential in STK2. Mochizuki and Nakahara [35] employed a xenogeneic serum-free culture medium (XFM) (containing a basal medium of amino acids, vitamins, inorganic salts, and glucose, along with a supplement that replaces animal serum with human or chemically defined alternatives, growth factors, hormones, and lipids) for handling DPSCs. The DPSCs cultivated in XFM exhibited typical MSC characteristics in vitro and in vivo, including marker gene/protein expression, trilineage differentiation potential, and hard osteo-dentin tissue formation. Xiao et al. [36] established a chemically defined serum-free culture system [called serum-free Essential 8 medium (E8); provided by Thermo Fisher Scientific Co., Ltd., Waltham, MA, USA] for culturing DPSCs. Compared with serum-containing medium, E8 medium exhibited better ability to maintain cell proliferation, pluripotency, migration, and stability. However, some researchers indicated that these media do not adequately support DPSC proliferation and differentiation. On the other hand, various human blood derivatives have been proposed as alternatives to animal serum for stem cell culture, including autologous or allogenic human serum, human plasma, and human platelet lysates (PL) with their released factors. Marrazzo et al. [37] found that a low concentration of PL (1%) was able to support the growth and maintain the viability of DPSCs in vitro. Additionally, PL was shown to be a suitable option for protocols promoting osteogenic and chondrogenic differentiation of DPSCs.

### 2.4. Single-Cell Cloning and Single-Cell-Based RNA-Seq (scRNA-Seq)

Gronthos et al. [11] first demonstrated that human DPSCs can be isolated through DPSC-ED. The same group [38] also attempted to obtain single-colony-derived DPSC clones to determine whether DPSCs have odontogenic potential. Of the 12 individual clones analyzed, two-thirds generated abundant ectopic dentin in vivo, while only a limited amount of dentin was detected in the remaining one-third. These results suggest that DPCs are composed of various types of DPSCs.

RNA-seq analysis is a useful tool to distinguish the properties of cell types. This approach is particularly powerful for identifying cells at the single-cell level through transcript quantification in individual cells [39]. Kobayashi et al. [40] examined the transcriptome to identify new candidate marker genes in DPSCs using RNA-seq analysis of clonally isolated cells from heterogeneous multipotent human DPSCs. They first obtained 50 colony-forming single-cell-derived clones from an 11-year-old female patient. DNA microarray analysis of five representative clones with strong differentiation potentials revealed 1227 genes related to multipotency. Ninety of these genes overlapped with those involved in “stemness or differentiation”. Based on the predicted locations of expressed protein products, 14 of the 90 genes were selected as candidate DPSC markers, which are particularly associated with multipotential characteristics. These included desmoplakin (*DSP*), serpin family E member 1 (*SERPINE1*), sortilin 1 (*SORT1*), collagen type I alpha 2 chain (*COL1A2*), collagen type III alpha 1 chain (*COL3A1*), ATPase phospholipid transporting 8B1 (*ATP8B1*), intercellular adhesion molecule-1 (*ICAM1*), adhesion G protein-coupled receptor A2 (*ADGRA2*), anthrax toxin receptor 1 (*ANTXR1*), oxytocin receptor (*OXTR*), and serglycin (*SRGN*). These new candidate marker genes may be useful for analyzing or enriching multipotent stem cells.

### 2.5. Usefulness of mRNA Analysis of a Small Number of DPSCs for Creating Gene Expression Profiles

Although scRNA-seq is currently the gold standard for analyzing cell-specific expression profiles, one major limitation is the requirement for expensive, cell-sorting equipment (FACS) to exclude debris and dead or unwanted cells before sequencing. Furthermore, this approach requires pre-staining of live cells with antibodies recognizing cell surface antigens. Inada et al. [41] developed a novel method enabling mRNA analysis of a small number of fixed and immuno-stained cells, specifically expressing octamer-binding transcription factor 3/4 (OCT3/4). OCT3/4 is a transcriptional factor essential for maintaining stemness in various stem cells and is expressed in only a few HDDPCs [42]. Since OCT3/4 is localized in the nucleus (and not on the cell surface), it cannot be used with FACS, which relies largely on the isolation of live cells using cell surface antigen markers. However, isolation of 3,3′-diaminobenzidine-stained *OCT3/4*-expressing cells is possible by manually collecting immuno-stained cells with a mouth-controlled micropipette under a dissecting microscope. When these manually isolated cells (2~10) were subjected to RNA isolation using a whole transcriptome amplification (WTA) kit, successful amplification of *OCT3/4* mRNA-derived RT-PCR products was achieved [41]. This approach is termed “RNA analysis based on a small number of manually isolated fixed cells (RNA-snMIFxC)”. Notably, a similar approach was reported by Mutisheva et al. [43], who attempted to establish a method for FACS sorting of fixed cells for scRNA-seq without compromising data. Using murine pancreatic ductal adenocarcinoma samples, in which the number of live cells is limited, they compared semi-automated mechanical/enzymatic digestion of solid tumors using the gentleMACS Dissociator with mechanical dissociation alone. This protocol enabled tissue dissociation and staining within a single day, followed by subsequent cell sorting and scRNA-seq, which represents a major advantage for processing clinical patient material.

### 2.6. Markers for DPSCs

As previously shown by Gornthos et al. [38] and through RNA seq analysis [44], DPSCs are a heterogeneous mixture of cell populations. This heterogeneity is further supported by the expression of stemness markers, such as OCT3/4 and SRY-related HMG-box 2 (SOX2), which are predominantly expressed in the nucleus. For example, immunocytochemical staining of HDDPC-derived DPSCs demonstrated that some cells expressed these markers, whereas others did not [42]. Notably, Kawashima et al. [45] listed CD13 (aminopeptidase N), CD146 (melanoma cell adhesion molecule; MCAM), and CD166 (activated leucocyte cell adhesion molecule; ALCAM) as new cell surface markers for DPSCs, as well as CD29 and CD44, both of which are already shown in Table 1. Furthermore, Madhoun et al. [14] reported that DPSCs express several neural stem cell markers, including nestin (NES), vimentin (VIM), β-III tubulin (TUBB3), musashi1 (MSI1), galactosylceramidase (GALC), S100 calcium-binding protein B (S100B), neurofilament heavy chain (NEFH), S100 calcium-binding proteins (S100), Notch1 (NOTCH1), CD271 (nerve growth factor receptor; NGFR), and synaptophysin (SYP).

### 2.7. Multilineage Differentiation Potential of DPSCs

The plasticity of DPSCs allows them to differentiate into various cell types of endodermal (respiratory and gastrointestinal tracts, liver, pancreas, thyroid, prostate, and bladder lineages), mesodermal (adipogenic, osteogenic, and chondrogenic lineages), and ectodermal (skin and neural lineages) origin [46]. By manipulating culture conditions, specific differentiation pathways can be restricted, enabling the generation of cultures enriched in particular precursors in vitro. Table 2 lists experiments on differentiation induction of human DPSCs using non-gene-engineered approaches.

#### 2.7.1. Osteogenic (Osteoblastic) Differentiation

Osteogenic (osteoblastic) differentiation of DPSCs has been successfully induced using chemical reagents such as dexamethasone, L-ascorbic acid, and β-glycerol phosphate supplementation [49,52]. In addition to cultivation in osteogenic medium, various approaches have been reported for inducing differentiation into the osteogenic lineage, including co-cultivation of DPSCs with endothelial cells [47], cultivation in the presence of human serum [48], concentrated growth factor exudate (CGFe), and transforming growth factor beta 1 (TGF-β1) [54], or betaine (BET) [55]. Growth of DPSCs in the presence of various scaffolds [50,57,58], decellularized adipose tissue solid foams [56], or microcapsules [53] has also resulted in successful osteogenic differentiation. Alizarin red S staining is typically used to confirm matrix mineralization and calcium deposition after induction [51]. Concomitantly, expression of bone-specific proteins such as alkaline phosphatase (ALP), collagen type I (COL1), osteocalcin (OCN), osteonectin (ON), osteopontin (OPN), osterix (Osx), and runt-related transcription factor 2 (RUNX2) is also detected [49].

#### 2.7.2. Odontoblastic Differentiation

For odontoblastic differentiation of human DPSCs, Baldión et al. [64] used odontogenic induction medium (containing TGF-β1, dexamethasone, β-glycerophosphate, and ascorbic acid) and obtained odontoblast-like cells with odontoblast markers such as dentinal sialophosphoprotein (DSPP), dentin matrix protein-1 (DMP-1), and RUNX2 transcripts. These cells exhibited mineral deposition activity, as evidenced by Alizarin Red S and von Kossa staining, along with low ALP activity. Odontogenic differentiation was also achieved by cultivating DPSCs on dentin [59], calcium silicate materials [61], or scaffolds [62,63]. Furthermore, cultivation of DPSCs in preameloblast-conditioned medium (PA-CM) [60] with endothelin-1 (ET-1) [65] or vascular endothelial growth factor A (VEGFA) [66] also led to the generation of odontoblast-like cells.

#### 2.7.3. Differentiation into Adipogenic Cell Lineages

For differentiation into adipogenic cell lineages, DPSCs are treated with insulin, dexamethasone, indomethacin, and 3-isobutyl-1-methylxanthine (IBMX) [52,69]. Enhanced adipogenic differentiation of DPSCs has also been observed in enzymatically decellularized adipose tissue solid foams [70]. The resulting adipogenic cells contain lipid droplets in their cytoplasm, which can be visualized using Oil Red O staining [67]. The adipogenic phenotype is usually further confirmed by the expression of several adipogenic markers, such as peroxisome proliferator-activated receptor γ (PPAR-γ), glucose transporter type 4 (GLUT4), fatty acid binding protein 4 (FABP4), and lipoprotein lipase (LPL) [68].

#### 2.7.4. Differentiation into Neurogenic Cell Lineage

DPSCs have been shown to differentiate into neuron-like cells, dopaminergic neurons, oligodendrocytes, and Schwann cells when cultivated in specific media that promote neuronal differentiation [78]. For example, Osathanon et al. [72] demonstrated that neurogenic differentiation was achieved when DPSCs were first cultured in neuroinduction medium containing B27 (a serum-free supplement optimized to promote neuronal cell differentiation), L-glutamine, bFGF, and epidermal growth factor (EGF) for 7 days. Subsequently, single cells derived from spheres were seeded onto collagen type IV-coated dishes in neuroinduction medium supplemented with retinoic acid for another 7 days. DPSCs have also been induced into neuron-like cells after incubation in commercially available neurogenic differentiation medium [73], conditioned medium of cerebrospinal fluid and retinoic acid (RA) [74], or medium containing nerve growth factor (NGF) [76]. Culturing in the presence of graphene–polycaprolactone hybrid nanofibers [75] or under high K^+^ stimulation (50 mM KCl) [77] can also induce transdifferentiation of DPSCs into neuron-like cells. Successful neurogenic differentiation is confirmed by the expression of several neural markers, including neuronal nuclei (NeuN), microtubule-associated protein 2 (MAP2), neural cell adhesion molecule (NCAM), growth-associated protein 43 (GAP43), glial fibrillary acid protein (GFAP), synapsin I (SYN1), and TUBB3 [96].

#### 2.7.5. Differentiation into Hepatocytes

Hepatic differentiation of human DPSCs has been successfully demonstrated by incubating CD117-positive DPSCs in medium supplemented with hepatic growth factor, insulin-transferrin-selenium-x, dexamethasone, and oncostatin M [79]. The resulting hepatocyte-like cells exhibited substantial cytoplasmic glycogen storage and expressed hepatic markers such as alpha-fetoprotein (AFP), albumin, hepatic nuclear factor-4 alpha (HNF4α), insulin-like growth factor-1 (IGF-1), and carbamoyl phosphate synthetase (CPS) [79].

#### 2.7.6. Differentiation into Endothelial Lineage

Endothelial differentiation (vasculogenesis) can be achieved by incubating DPSCs in a medium supplemented with a mixture of B27, heparin, and growth factors, including VEGF-A165 [83]. In particular, in vitro Matrigel assays demonstrated that DPSCs can generate vascular tubules in 3D cultures. Bento et al. [80] reported that DPSCs exposed to endothelial growth medium supplemented with VEGF differentiated into VEGF receptor 2 (VEGFR2)-positive and CD31-positive endothelial cells in vitro. They suggested that the VEGF/mitogen-activated protein kinase kinase 1 (MEK1)/extracellular signal-regulated kinase (ERK) signaling pathway acts as a key regulator of DPSC endothelial differentiation. Furthermore, co-cultivation with human umbilical vein endothelial cells (HUVECs), after encapsulation in a scaffold system of self-assembling peptide nanofibers, resulted in the generation of DPSC-derived endothelial cells [82]. Endothelial differentiation can be confirmed by the expression of several angiogenesis-related markers, including CD54/ICAM-1, CD146/MCAM, monocyte chemotactic protein-1 (MCP-1), von-Willebrand factor (VWF) (domain 1 and 2), CD31/platelet/endothelial cell adhesion molecule-1 (PECAM-1), VEGF, CD34, fetal liver kinase 1 (Flk-1)/VEGFR2, bFGF, insulin-like growth factor binding protein-3 (IGFBP3), interleukin-8 (IL-8), plasminogen activator inhibitor-1 (PAI-1), tissue inhibitors of metalloproteinase-1 (TIMP-1), and urokinase-type plasminogen activator (uPA) [81].

#### 2.7.7. Differentiation into Cardiomyocytes

DPSCs can differentiate into cardiomyocytes when incubated in the presence of 5-azacytidine, a well-known demethylating agent, for 2 days [84]. The resulting embryoid bodies (EBs) express various precardiac markers, including myosin heavy chain 6 (MYH6) and mesoderm posterior BHLH transcription factor 1 (MESP), as well as markers for the three germ layers. When these EBs are plated onto gelatin-coated tissue culture dishes, they develop functional cardiomyocytes expressing cardiac markers such as NK-2 transcription factor related, locus 5 (Nkx2.5), and connexin 43 (*Cx43*).

#### 2.7.8. Differentiation into Pancreatic Lineage

Mesenchymal stem cells can be induced into the pancreatic cell lineage using published three-step induction protocols designed to enhance pancreatic progenitor commitment and production yield. These protocols often employ growth factors, such as activin A and retinoic acid, along with other factors, like nicotinamide and laminin, to guide cell differentiation [86]. Ishkitiev et al. [86] confirmed that CD117^+^ DPSCs (which express CD117, a receptor tyrosine kinase encoded by the KIT gene) can differentiate into the pancreatic lineage when subjected to the three-step induction protocol. The resulting cells expressed pancreatic-specific endocrine markers, including insulin, glucagon (GCG), somatostatin (SST), pancreatic polypeptide (PPY), and exocrine marker amylase-2a (AMY2A). Additional markers such as pancreatic and duodenal homeobox 1 (PDX1), hematopoietically expressed homeobox (HHEX), motor neuron and pancreas homeobox 1 (MNX1), neurogenin 3 (NEUROG3), paired box 4 (PAX4), paired box 6 (PAX6), and NK6 homeobox 1 (NKX6-1) were also detected. Similar findings were reported by several other laboratories [85,89,90,91]. Notably, Abuarqoub et al. [92] employed PSC Definitive Endoderm Induction media (Thermo Fisher Scientific) to induce pancreatic β cell differentiation.

Islets containing insulin-producing cells (IPCs) have also been successfully generated using DPSCs. Kanafi et al. [87] used a stepwise protocol to accelerate differentiation towards the pancreatic lineage and succeeded in generating islet-like cell clusters (ICCs). These ICCs were embedded in immuno-isolatory biocompatible macro-capsules and into streptozotocin (STZ)-treated diabetic mice. The grafted mice exhibited normoglycemia 3–4 weeks after transplantation, providing an autologous and non-controversial source of human tissue for potential stem cell therapy in diabetes. Mendoza et al. [88] further demonstrated that cultivating DPSCs in a 3D culture system using a stepwise protocol generated organoid-like 3D structures resembling pancreatic islets. These structures showed enhanced insulin and C-peptide production, expressed multiple pancreatic markers, and exhibited glucose-dependent insulin secretion.

#### 2.7.9. Differentiation into Smooth Muscle Cells (SMCs)

Song et al. [93] incubated DPSCs in bladder SMC-conditioned medium supplemented with TGF-*β*1 for 2 weeks. The resulting cells expressed SMC-specific markers, including alpha smooth muscle actin (*α*-SMA), desmin (DES), and calponin (CNN), along with myosin. Xu et al. [94] used TGF-β1 and bone morphogenetic protein 4 (BMP4) to test the induction of DPSCs into SMCs and concluded that TGF-β1, but not BMP4, is effective for that purpose. Zhang et al. [95] demonstrated that direct contact with endothelial cells in the presence of TGF-β1 is required to drive DPSCs toward SMC differentiation. Song et al. [93] suggest that human DPSCs could serve as an alternative and less invasive source of stem cells for smooth muscle regeneration.

## 3. Gene Engineering of DPSCs

DPSCs have shown great potential as a cell-based therapy for regenerative medicine since they can be isolated relatively easily and possess properties similar to those of bone marrow-derived MSCs. However, their utilization alone is limited due to the insufficient number of available cells and the low production of therapeutic factors needed to repair injured sites after transplantation. One approach to overcome these limitations is to generate gene-engineered DPSCs, enabling enhanced cellular differentiation (including transdifferentiation) into various cell types, production of cells that secrete specific therapeutic proteins, and generation of immature cells with high pluripotency, as exemplified by iPSCs. These genetically modified (GM) DPSCs hold promise for a wide range of tissue engineering applications as a cell-based therapy. In Figure 2, a summary of recent achievements in gene-engineered DPSCs is shown.

### 3.1. Development of Methods for Gene Introduction into DPSCs

To date, several methods have been developed for gene delivery into DPSCs, including transduction using lentiviral vectors [97,98,99,100,101,102], adenoviral vectors [103], retroviral vectors [104], and electroporation or chemical reagent-based transfection using plasmids or transposons [105,106,107,108,109,110,111]. In particular, “loss-of-function” experiments using RNA interference (RNAi)-based approaches, such as using siRNA and shRNA, as well as microRNA (miRs), have been widely applied. MiRs directly or indirectly regulate in vitro DPSC differentiation into various cell types and functions through different signaling pathways [112]. For example, Liu et al. [113] first applied an miR-based strategy to promote odontoblastic differentiation of mouse DPSCs and reported increased expression of DSPP and DMP-1 following lentivirus-mediated introduction of miR-145 and miR-143. Mechanistically, MiR-145 binds to the 3′-UTRs of Krüppel-like factor 4 (KLF4) and OSX, suppressing their expression. Downregulation of miR-143 partially reduces miR-145 levels, thereby releasing KLF4 and OSX expression. Upregulation of these transcription factors, in turn, enhances the expression of the odontoblast markers DSPP and DMP-1, ultimately driving odontoblast differentiation.

### 3.2. Immortalization

Primary cells, including DPSCs, have limited growth potential due to cellular senescence. Immortalization is a technique by which primary cells acquire continuous proliferative capacity without losing the characteristics of their parental cells [114]. Immortalization can be achieved through gene delivery of human telomerase reverse transcriptase protein (hTERT), E6/E7 proteins from human papilloma virus 16 (HPV16), or simian virus 40 large T antigen (SV40T) [115]. Inada et al. [107] immortalized HDDPCs by *piggy*Bac-based transfection with hTERT and HPV16 genes. The resulting immortalized cells demonstrated continuous proliferation but lacked tumorigenic potential, as confirmed by transplantation under the skin of immunocompromised mice. Moreover, they retained the properties of parental DPSCs, including mineralizing capability, expression of specific markers, and multipotency. These immortalized DPSCs are now recognized as a valuable cell model for studying the mechanisms of odontoblast proliferation and differentiation [116]. Zanini et al. [117] showed that Biodentine, a new tricalcium silicate-based cement, induced differentiation of immortalized murine DPSCs into odontoblast-like cells and enhanced both cell proliferation and biomineralization. Similarly, betaine (BET) treatment promotes osteogenic differentiation in immortalized or primary-cultured human DPSCs, accompanied by increased mineral deposition and upregulation of osteogenic marker expression [52,118].

### 3.3. Alteration of Cell Behavior

Using these gene delivery methods, several biologically important subjects have been investigated. Recent reports on altered cell behavior and properties following gene delivery are summarized in Table 3.

#### 3.3.1. Enhanced Differentiation into Odontoblast/Mineralization

Gene delivery experiments using DPSCs have demonstrated that several molecules enhance their differentiation potential towards odontoblasts. These include growth/differentiation factor 11 (Gdf11) [105], KLF4 [120], DNA damage-inducible transcript 3 (DDIT3) [124], inhibitor of DNA binding 1 (ID1) [125], zinc finger and BTB domain-containing 20 (ZBTB20) [126], SRY-related HMG-box 2 (SOX2) [128], platelet-derived growth factor-BB (PDGF-BB) [99], DMP-1 [134], and long noncoding RNA H19 (LncRNA H19) [135]. Notably, overexpression of the Notch ligand Jagged-1 (JAG-1) activated the Notch signaling pathway, inhibited odontoblastic differentiation, and mineralized tissue formation [119]. These results highlight the role of Notch signaling in regulating DPSC behavior. Wang et al. [121] used lentivirus-mediated knockdown for the Notch ligand Delta1 (also called delta-like canonical Notch ligand, DLL1) in DPSCs. DLL1-deficient DPSCs exhibited accelerated differentiation into odontoblasts. Furthermore, Chen et al. [123] showed that transfection with RNAi for CD44, a hyaluronan-binding cell surface signal transducing receptor, suppressed mineralization, suggesting a role of CD44 in DPSC mineralization.

#### 3.3.2. Enhanced Differentiation into Osteogenic Lineage

Since DPSCs are considered a promising source for tissue engineering, particularly for osteogenic tissues, increasing their osteogenic capacity is crucial for their potential application in tissue engineering. Therefore, it is important to identify new therapeutic targets and elucidate the mechanisms regulating the osteogenic differentiation of DPSCs [177]. Several molecules promote osteogenic differentiation in DPSCs, including B cell lymphoma-2 (BCL-2) [127], SOX2 [138], TGF-*β*1 [111], Wnt family member 4 (WNT4) [139], ephrinB2 (EFNB2) [101], hsa_circ_0026827 [140], hypoxia-inducible factor-1α (HIF-1α) [141], and semaphorin 3A (SEMA3A) [142]. Growing evidence indicates that miRs influence osteogenic differentiation of DPSCs by regulating different aspects of this process. For example, Zhang et al. [178] reported that overexpression of miR-143, which is involved in metabolism, cell proliferation, inflammation, and apoptosis, inhibited osteogenic differentiation of DPSCs. Bioinformatics analysis and luciferase reporter assays showed that tumor necrosis factor-alpha (TNF-α) is a direct target of miR-143 in DPSCs, with miR-143 regulating TNF-α expression through post-transcriptional binding to its 3′UTR. Functional analysis further revealed that miR-143 inhibited TNF-α-induced osteogenic differentiation of DPSCs, suggesting that miR-143 suppresses this process by downregulating TNF-α. Conversely, inhibition of miR-143 promoted osteogenic differentiation of DPSCs through activation of the nuclear factor kappa B (NF-κB) signaling pathway.

#### 3.3.3. Enhanced Cell Proliferation

Cell proliferation of DPSCs is promoted by several factors, including overexpression of DLL1 [121], DDIT3 [124], SOX2 [145], PDGF-BB [99], stromal cell-derived factor-1α (SDF-1α) [148], VEGF [148], Lin28 [149], TGF-β1 [111], LncRNA H19 [153], and adrenomedullin (ADM) [154]. RNAi-mediated suppression of *DLL1* in DPSCs resulted in the inhibition of proliferation [121], suggesting the involvement of the Notch–Delta signaling pathway in this process. The external environment is also known to affect DPSC proliferation. Fukuyama et al. [144] demonstrated that siRNA-mediated downregulation of the catalytic alpha 1 subunit of AMP-activated protein kinase (AMPK) (AMPKα1)—a stress-responsive enzyme activated by hypoxia—led to decreased proliferation under both normoxic and hypoxic conditions, indicating that AMPK plays an important role in DPSC’s reactions to hypoxia, presumably in maintaining energy homeostasis. According to Fukuyama et al. [144], hypoxia leads to the activation of AMPK via an increase in the AMP/ATP ratio, and AMPK acts as an intercellular energy sensor, maintaining energy balance during hypoxia in a variety of cells. Consistent with this, miR-210-3p, a major hypoxia-inducible miR (also known as hypoxamir, widely expressed in many cell types), promoted proliferation and early differentiation of DPSCs under hypoxia [150]. Hara et al. [122] showed that transfection of miR-720, which targets the stem cell marker *NANOG*, reduced DPSC proliferation while promoting odontogenic differentiation through *NANOG* suppression. *Similarly*, Zhang et al. [150] reported that overexpression of miR-633, which targets the 3′ UTR of matrix extracellular phosphoglycoprotein (*MEPE*), significantly enhanced both proliferation and differentiation of DPSCs. They concluded that MEPE is a molecular biomarker regulating the differentiation of DPSCs.

#### 3.3.4. Enhanced Angiogenic Commitment

DPSCs play a critical role in angiogenesis during dental pulp tissue repair and regeneration. The angiogenic capacity of DPSCs can be enhanced through overexpression of angiogenic factors such as PDGF-BB [99,158], SDF-1α or VEGF [148], BCL-2 [156], and Ets variant transcription factor 2 (ETV2) [157]. For example, Zhu et al. [148] demonstrated that overexpression of SDF-1α or VEGF caused enhanced cell proliferation and endothelial cell migration. Notably, culturing these engineered cells in Matrigel led to vascular-tube formation. These results suggested that a combination of VEGF-overexpressing and SDF-1α-overexpressing DPSCs could further increase the extent of vascularized dental pulp regeneration in vivo.

miRs are also known to indirectly regulate DPSC differentiation into vascular endothelial cells by modulating factors, such as hypoxia-inducible factor-1 (HIF-1) and VEGF. For instance, Liu et al. [155] identified VEGF and kinase insert domain-containing receptor (KDR) as targets of miR-424 during endothelial differentiation of DPSCs. Inhibition of miR-424 promoted endothelial differentiation, whereas its overexpression suppressed angiogenic potential. Specifically, miR-424 inhibition enhanced the secretion of angiogenic factors and upregulated receptor expression, thereby improving differentiation efficiency.

#### 3.3.5. Enhanced Neurogenic Differentiation and Neuroprotective Effects

DPSCs can be differentiated into neural cells (neurons and glia) in vitro using growth factors such as brain-derived neurotrophic factor (BDNF), glial cell-derived neurotrophic factor (GDNF), bFGF, and RA in specific induction media. The resulting neural cells can potentially be used for therapeutic treatment of neurodegenerative diseases. Differentiation of DPSCs into neural cells via gene delivery of exogenous DNA, which reprograms the cells towards a neuronal lineage, is also possible. For example, Behrouznezhad et al. [159] hypothesized that overexpression of zinc finger protein 521 (Zfp521), a transcription factor regulating several genes, in DPSCs could facilitate neural differentiation through chromatin modifications that activate neural specification genes.

A reliable method for converting DPSCs into neuronal stem cells (NSCs) has not yet been fully established. Gancheva et al. [161] employed OCT3/4 overexpression in combination with neural inductive conditions to reprogram DPSCs into NSCs. The reprogrammed DPSCs exhibited a neuronal-like phenotype, increased neural marker expression, limited sphere-forming ability, and enhanced neuronal but not glial differentiation. In vivo analysis using a developmental avian model showed that implanted DPSCs survived in the developing central nervous system (CNS) and responded to endogenous signals, displaying neuronal phenotypes. These findings indicate that OCT3/4 enhances the neural potential of DPSCs, confirming characteristics consistent with neuronal progenitors.

#### 3.3.6. Enhanced Adipogenic Commitment

RNA-seq analysis demonstrated that several adipogenesis-related genes are influenced by complex methylation changes, leading to a differential gene expression profile in adipocytes under lean and obese conditions [179]. Argaez-Sosa et al. [180] showed that the ability of DPSCs to differentiate into adipocytes may be linked to the expression patterns of DNA methylation-related genes acquired during adipogenic induction. They again [162] showed that overexpression of ten-eleven-translocation 2 (TET2), a key regulator of DNA methylation during adipogenic induction, in DPSCs increased the expression of adipogenic marker genes and enhanced the transition of DPSCs toward adipogenic commitment.

#### 3.3.7. Reduced Inflammatory Responses

Hepatocyte growth factor (HGF) exhibits both pro-inflammatory and anti-inflammatory effects in the context of DPSCs and inflammation, depending on the stimulus [181]. Meng et al. [164] demonstrated that overexpression of HGF in DPSCs led to reduced spleen mass in psoriatic mice and downregulation of inflammation-related factors, such as interferon-gamma (IFN-γ), TNF-α, and interleukin (IL)-17A, suggesting that DPSCs exert enhanced therapeutic effects on psoriasis by reducing inflammatory responses. Furthermore, Ni et al. [165] demonstrated that overexpression of WNT4 alleviated the inflammatory response, enhanced BCL-2 expression, and decreased apoptosis rate in DPSCs stimulated with lipopolysaccharide (LPS) by inhibiting the inhibitor of κB kinase (IKK)/nuclear factor-kappa B (NF-κB) pathway.

#### 3.3.8. Enhanced Cell Migration

Peptidylprolyl cis/trans isomerase, NIMA-interacting 1 (PIN1), a member of the prolyl isomerase family, specifically binds to phosphorylated Ser/Thr-pro motifs to catalytically regulate the post-phosphorylation conformation of its substrates. Kim et al. [166] investigated the role of Pin1 expression in DPSCs in regulating the P2Y1 receptor, a purinergic G protein-coupled receptor (GPCR) activated by ADP, and activating ADP-mediated P2Y1 signaling. Silencing *PIN1* by siRNA reduced the migration of DPSCs expressing the P2Y1 receptor, suggesting that PIN1 is involved in P2Y1-mediated cell migration.

#### 3.3.9. Generation of IPCs and Oligodendrocyte Progenitors (OPs) from DPSCs by Forced Expression of Exogenous Genes

In 2008, Melton and colleagues [182] first demonstrated that three transcription factors—neurogenin 3 (*Ngn3*; also known as *Neurog3*), *Pdx1*, and MAF bZIP transcription factor A (*Mafa*)—introduced in vivo could successfully reprogram differentiated pancreatic exocrine cells in adult mice into cells closely resembling β cells. This process is referred to as “direct reprogramming of hepatocytes to pancreatic cells”. The induced β cells were indistinguishable from endogenous islet β cells in size and shape, expressed genes essential for β cell function, and were able to ameliorate hyperglycemia in diabetic animal models. Since then, successful conversion of non-β cells, such as DPSCs [167,168] and keratinocytes [183] into β cells, has been reported, suggesting that reprogramming into β cells is not strictly limited to lineage-related cells. For example, Nozaki and Ohura [169] reported that *miR-183* was downregulated during the direct conversion of DPSCs to IPCs based on miR array profiling. The *miR-183* family, which includes miR-183, miR-96, and miR-182, is highly expressed in a variety of non-sensory diseases, including cancer, neurological, and autoimmune disorders [184]. Nozaki and Ohura [168] further demonstrated that downregulation of *miR-183* in DPSCs resulted in the generation of insulin-expressing cells within 72 h after ectopic expression of an *miR-183* inhibitor.

This concept broadens the potential gene-based transdifferentiation approaches in regenerative medicine, as various cells of different origins may serve as candidates. For example, the ability of DPSCs to transdifferentiate into OP was demonstrated by Askari et al. [170], who transfected DPSCs with the human oligodendrocyte transcription factor 2 (*Olig2*) gene. Olig2 encodes a helix–loop–helix transcription factor essential for lineage determination of oligodendrocytes and serves as an inducer for the oligodendrogenic pathway. Following differentiation, the resulting cells expressed several oligodendrogenic markers and were identified as OP cells. Notably, transplantation of these OPs into a mouse model of local sciatic nerve demyelination (induced by lysolecithin) resulted in active remyelination. These findings suggest that DPSCs can be programmed into oligodendrocyte progenitors after forced expression of *Olig2*, highlighting their potential as a simple and valuable source for cell therapy of neurodegenerative diseases.

#### 3.3.10. Apoptosis

The apoptotic mechanism underlying the development of DPSCs remains unclear. To investigate this, knockdown experiments using an RNAi approach were performed. Knockdown of caspase-9 (CASP9) in DPSCs resulted in a significant reduction in apoptosis, which was associated with increased caspase-3 (CASP3) activity [171]. These results indicate that CASP9 and activated CASP3 predominantly regulate cell apoptosis in DPSCs.

miR-224-5p functions as an endogenous inhibitor of specific target genes and plays diverse roles in biological processes, including cancer cell proliferation, invasion, and metastasis [185]. In DPSCs, miR-224-5p protects cells against apoptosis by downregulating Ras-related C3 botulinum toxin substrate 1 (Rac1), a member of the Rac family of guanosine triphosphate phosphohydrolases [172]. Similar protective effects have been observed in H9C2 cardiomyocytes, a well-known immortalized rat embryonic heart-derived cell line. Li et al. [186] reported that miR-194-5p attenuated the accumulation of reactive oxygen species (ROS) and alleviated hypoxia/reoxygenation (H/R)-induced apoptosis in H9C2 cardiomyocytes.

#### 3.3.11. Skeletal Myogenic Differentiation

miRs can promote myogenic differentiation of DPSCs. For example, Li et al. [173] first demonstrated that human adult DPSCs treated with 5-aza-2′-deoxycytidine exhibited myogenic differentiation in vitro, a process during which the expression of miR-135 and miR-143 was markedly decreased. miR-135 shows brain-specific expression and is involved in neuronal morphology, neural induction, and synaptic function [187], while miR-143 plays a role in the differentiation of stem cells into various lineages, including smooth muscle cells and adipocytes [188]. Based on these findings, Li et al. [173] co-transfected DPSCs with antisense oligonucleotides against miR-143 and miR-135. This treatment led to approximately half of the cells exhibiting distinct cardiomyocyte-like characteristics, with expression of myogenic markers such as myocyte enhancer factor 2C (MEF2C), myogenic differentiation 1 (MyoD), myogenin (MyoG), and myosin heavy chain (MyHC). These findings suggest that miRs play a decisive role in inducing the myogenic differentiation of DPSCs and provide new insights into their use for muscle regeneration therapies. Furthermore, miR-139-5p, which functions as a tumor suppressor by regulating gene expression and inhibiting cancer growth and spread [189], has also been implicated in myogenesis. Overexpression of miR-139-5p in DPSCs activated the Wnt/β-catenin signaling pathway, promoted cell growth, and induced skeletal myogenic differentiation [174]. Conversely, inhibition of miR-139-5p suppressed the Wnt/β-catenin signaling pathway, resulting in reduced cell growth and impaired skeletal myogenic differentiation in DPSCs.

#### 3.3.12. Enhanced Pluripotency and Multilineage Differentiation Capability

OCT3/4 plays a critical role in early mammalian development by regulating the formation and maintenance of iPSCs/ESCs and the inner cell mass of the blastocyst. OCT3/4 exerts its effects by binding to DNA and often cooperates with other transcription factors, such as SOX2 and NANOG, to control gene expression and preserve the undifferentiated state of pluripotent cells [190]. It has also been reported to act as a pioneer transcription factor, with subsequent factors and environmental cues directing cell fate [191,192]. With regard to DPSCs, limited experimental evidence exists on their transcription factor-mediated reprogramming. Liu et al. [175] demonstrated that induced overexpression of OCT4A (the OCT3/4 isoform considered the key regulator of pluripotency) in DPSCs enhanced their differentiation capacity toward neural and glial lineages while also promoting the expression of other pluripotency regulators. These findings suggest a potential role for OCT4A in facilitating multipotency rather than strictly maintaining pluripotency. In other words, expression levels are critically important. Sufficient levels maintain the pluripotent state, while excessive overexpression can initiate differentiation into specific cell types.

Xeroderma pigmentosum complementation group C protein (XPC), a component of the DNA repair pathway, also interacts with OCT3/4 to modulate pluripotency. Liu et al. [176] demonstrated that overexpression of XPC in DPSCs led to significantly higher expression of the stemness markers OCT3/4, SOX2, and c-MYC. Furthermore, this treatment enhanced proliferation, reduced apoptosis, and improved multilineage differentiation, suggesting that XPC plays a crucial role in maintaining or enhancing the pluripotency of DPSCs.

### 3.4. Critical Evaluation of Gene-Engineering Strategies

Signaling pathways, such as Wnt/β-catenin, TGF-β/SMAD, and Notch, are critical regulators of DPSC behavior, coordinating self-renewal and differentiation for regenerative dentistry. This network of pathways controls the repair and regeneration of dentin and pulp tissue following injury or disease [193]. Gene-engineering strategies modulate these pathways to enhance the potential of DPSCs. For example, overexpression of SOX2 resulted in enhanced odontoblast differentiation with expression of odontoblast markers, such a DMP-1. This event is closely associated with the activation of the Wnt signaling pathway [128]. PDGF-BB, a homodimer of the PDGF-B protein, which is known to be involved in cell proliferation, tissue repair, and angiogenesis, caused enhanced DPSC proliferation and odontoblastic differentiation. This event is carried out through activation of the phosphatidylinositol 3 kinase (PI3K)/protein kinase B (AKT) signaling pathway [99]. Overexpression of growth differentiation factor 15 (GDF15) caused enhanced osteogenic differentiation of DPSCs through activation of the TGF-β/SMAD signaling pathway, while knockdown of GDF15 produced the opposite effect [143].

## 4. Generation of Immature Cells by Forced Expression of Exogenous Genes in DPSCs

DPSCs are now known to be induced to generate iPSCs, as shown below. These cells can be obtained in large quantities in vitro, and many types of differentiated cells can subsequently be derived under specific induction conditions. In this context, iPSCs are widely considered a promising resource for regenerative medicine due to their ability to self-renew and differentiate into various cell types [4]. However, safety concerns are always associated with iPSCs when they are intended for use in regenerative medicine because iPSC-derived differentiated cells often include a mixture of immature cells (i.e., residual iPSCs) and differentiated cells. Notably, researchers have attempted to remove these immature cells after differentiation induction using chemicals, i.e., N-oleoyl serinol (S18), a ceramide analog [194], quercetin and YM155, inhibitors of survival [195], antibodies, such as keratan sulfate glycan-recognizing antibody [196], or a thymidine kinase-based cell ablation system [197], before utilizing them in regenerative medicine.

### 4.1. Generation of iPSCs

iPSCs can be generated from many types of dental cells, such as dental pulp, oral mucosa, gingiva, and periodontal ligament fibroblasts, making them a promising source for regenerative medicine in dentistry [198,199,200,201,202,203,204]. Tamaoki et al. [198] first attempted to isolate iPSC lines using DPSCs from young to old patients. They suggested that younger fibroblasts are more efficiently reprogrammed than older fibroblasts. Notably, non-juvenile cells, such as those from older adults (in which stem cells are not abundantly present), are generally more resistant to reprogramming with Yamanaka’s factors than juvenile cells because they have undergone more epigenetic changes associated with aging and lost part of their “reprogrammable” potential [205]. In agreement with this notion, Inada et al. [42] demonstrated that DPSCs with higher ALP activity tend to be more easily induced to generate iPSCs after transfection with Yamanaka’s factors, while DPSCs with no ALP activity failed to do so. Since ALP activity is closely associated with pluripotent cells, such as iPSCs and ESCs, DPSCs with higher ALP activity are believed to contain abundant stem cells. In other words, cells enriched with stem cells tend to be more easily induced to become iPSCs compared to those with fewer stem cells [206]. Supporting this hypothesis, repeated transfection of ALP-negative HDDPCs with Yamanaka’s factors resulted in increased endogenous ALP activity, together with acquisition of the ability to generate iPSCs [207]. These findings suggest that gene delivery of Yamanaka’s factors may increase the number of stem cells in ALP-negative cells, eventually leading to the generation of ALP-positive cells with the potential to form iPSCs.

Generally, human iPSCs and ESCs exhibit a more restricted differentiation capacity than their mouse counterparts, because they are derived from post-implantation epiblasts and represent a more committed, “primed” state of pluripotency [208]. These cells are called “epiblast stem cells (EpiSCs)” and have limited differentiation potential [209]. In contrast, mouse iPSCs and ESCs are referred to as “naïve stem cells” and have the ability to differentiate into various cells, including germ cells. Mouse iPSCs and ESCs (but not human counterparts) express a stem cell-specific antigen, stage-specific embryonic antigen-1 (SSEA-1), on their cell surface [210]. However, treatment of human iPSCs with drugs enabling conversion of EpiSCs to naïve stem cells for several days resulted in the generation of iPSCs with properties similar to naïve stem cells [211]. Concomitantly, the drug-treated human iPSCs began to express SSEA-1 [212]. Furthermore, these naïve stem cell-like human iPSCs exhibited broader differentiation potential, including the ability to form germ cells [212].

### 4.2. Generation of iTSCs from iPSCs

iTSCs are not fully reprogrammed to a pluripotent state like iPSCs but instead represent an intermediate stage between fully differentiated cells and pluripotent stem cells [4]. iTSCs have lost some of their original specialized functions but have not yet gained all the characteristics of pluripotency. They can be generated through partial reprogramming of various types of somatic cells, including hepatocytes, DPSCs, and pancreatic cells. To date, different types of iTSCs have been identified, including iTS-L [derived from liver cells (hepatocytes)] [213], iTS-P (derived from pancreatic cells) [214,215], and iTS-D (derived from DPSCs) [207]. iTSCs are considered to have potential applications in regenerative medicine due to their reduced tendency to form tumors (teratomas) compared to iPSCs [4,214]. More importantly, iTSCs express both stem cell and differentiated cell markers, as they are intermediate, partially reprogrammed cells. Furthermore, their marker profiles depend on the original tissue and the specific differentiation stage being captured. For example, iTS-P expresses stemness factors (*Oct3/4*, *Sox2*, *Klf4*, *Esg-1*, and *Rex1*) found in iPSCs/ESCs, together with *Pdx1*, which is not abundantly expressed in iPSCs/ESCs [214].

iTSCs can also be produced from iPSCs through cultivation under differentiation-inducing conditions. In this case, the state of human iPSCs [naïve stem cell or EpiSC state] is important for determining their fate. Kiyokawa et al. [216] demonstrated that EBs derived from naïve iPSCs—converted to a naïve state after pre-treatment with 2i (two small-molecule inhibitors, typically PD0325901 and CHIR99021)—were more efficiently induced to form pancreatic β-like cells after being subjected to an in vitro protocol compared with EBs derived from untreated iPSCs. These findings suggest that iTCS-P may be more abundantly produced from EBs derived from naïve iPSCs than from those derived from EpiSC-like iPSCs.

## 5. Usefulness of Feeder Cells to Maintain the Integrity of DPSCs

Feeder cells, exemplified by mitomycin-C (MMC)-treated mouse embryonic fibroblasts (MEFs), are mitotically inactivated (unable to divide) cells that provide essential nutrients, growth factors, and extracellular matrix components to support the survival and proliferation of target cells (such as stem cells) in a co-culture system. These feeder cells enable target cells to maintain their self-renewal and pluripotency by preventing spontaneous differentiation. They have been widely used for maintaining pluripotency in iPSCs and ESCs. Generally, culturing primary cells on plastic tissue culture dishes can often result in the loss of multipotency due to the inability of tissue-specific stem cells to survive in feeder-less conditions. Matrigel, a gelatinous protein mixture extracted from the Engelbreth–Holm–Swarm mouse sarcoma tumor, can be used as an alternative to feeder cells for maintaining stem cells [217], but this substance has significant limitations, including its undefined composition, variability between different production batches, and animal-derived origin. Notably, it was recently reported that culturing primary cells in medium containing feeder cells, particularly GM feeder cells expressing growth factors, may be beneficial for their survival and proliferation [218]. Ibanno et al. [219] demonstrated that GM human feeder cell-derived HDDPCs, engineered to express leukemia inhibitory factor (LIF), BMP4, and bFGF systemically, can be used to maintain the integrity of primary cultured HDDPCs. Co-culturing newly isolated HDDPCs with MMC-treated feeder cells results in enhanced cell proliferation of HDDPCs and a multipotent differentiation capability compared to HDDPCs cultured in the absence of feeder cells. Furthermore, HDDPCs co-cultivated with feeder cells continued to express stemness markers, while HDDPCs cultured in the absence of feeder cells failed to exhibit some of the stemness markers. These findings indicate that co-cultivation with feeder cells is beneficial for newly isolated HDDPCs to retain their stemness properties. However, preparing and maintaining feeder cells is a laborious task for researchers.

Notably, feeder cells can also remove contaminating bacteria (probably via phagocytosis) in a co-culture system. Saitoh et al. [220] demonstrated that STO feeder cells, which are also frequently used to maintain iPSCs and ESCs, are useful for the propagation of primarily cultured HDDPCs by eliminating contaminating bacteria and promoting cellular outgrowth.

## 6. Genome Editing

Periodontal disease and dental caries are common dental issues that can potentially be treated using the CRISPR/Cas9 genome editing system, a recently developed genetic engineering tool capable of cutting specific DNA sequences [221]. This system may enable personalized treatment and even the complete elimination of symptoms. The oral and craniofacial field harbors a wide range of diseases and developmental defects that require genetic models to fully exploit these genome editing techniques. For example, CRISPR/Cas9 can be used to disrupt dental-related gene functions or target *Staphylococcus aureus* (*S. aureus*) mutants, one of the major bacterial pathogens involved in tooth decay. Furthermore, genes associated with dental health—such as those involved in enamel formation or inflammatory response pathways—can be edited to reduce inflammation or genetic risk. However, the CRISPR/Cas9 system still presents challenges, particularly the risk of off-target effects, which may cause unintended gene expression and lead to occasional pathogenesis. Nonetheless, this technology represents an exciting advance that could revolutionize treatment strategies in dental health care [222,223]. In the following sections, recent reports on CRISPR/Cas9-based manipulation of virulent bacteria and genes relevant to dentistry are described.

### 6.1. CRISPR/Cas9-Based Manipulation of Bacterial Genes in S. aureus

Antibiotics target conserved bacterial cellular pathways or growth functions and, therefore, cannot selectively kill specific members of a complex microbial population. Bikard et al. [224] used the CRISPR/Cas9 system to destroy virulence-causing genes in pathogenic, but not non-pathogenic, *S. aureus*. Specifically, the Cas9/guide RNA (gRNA) complex targeted antibiotic resistance genes and successfully eliminated the *staphylococcal* plasmids carrying resistance determinants, thereby preventing the spread of plasmid-borne resistance. They also demonstrated that CRISPR/Cas9 antimicrobials function in vivo by killing *S. aureus* in a mouse skin colonization model. This technology opens opportunities to selectively manipulate complex bacterial populations as a new class of antimicrobials.

Gong et al. [225] developed a novel CRISPR/Cas9-based self-targeting gene editing strategy against glucosyltransferases (*gtfs*), major virulence factors involved in dental plaque biofilm. In bacteria, CRISPR/Cas9 relies on small RNAs (“spacers”) to restrict phage and plasmid infection, and it can also regulate endogenous gene expression using spacers targeting self-genes [226]. Gong et al. [225] designed self-targeting CRISPR arrays (with spacer sequences complementary to *gtfB*, which encodes glucosyltransferase B) and cloned them into plasmids. These plasmids were then transformed into *S. mutans* strain *UA159*—a bacterium isolated from a child with active caries and a well-recognized dental caries pathogen—as editing templates to generate desired mutants. As a result, this approach successfully edited *gtfB* and the *gtfB-gtfC* region, leading to disruption of biofilm formation.

### 6.2. CRISPR/Cas9-Based Manipulation of Genes Associated with Dental Health

Oxytocin (OT) is a neurohypophysial hormone that plays a role in lactation and parturition and exerts diverse biological actions via the OT receptor. Recently, OT has been suggested to promote bone formation by osteoblasts in osteoporosis. Kato and Yokose [227] hypothesized that OT can stimulate the differentiation of odontoblasts as well as osteoblasts. To test this, the *Oxtr* gene was knocked out by CRISPR/Cas9 in rat DPSCs. When intact rat DPSCs were treated with OT, mineralized nodule formation was induced along with increased mRNA expression of dentin sialoprotein and bone Gla protein. In contrast, *Otr* knockout (KO) cells showed inhibition of nodule formation and mRNA expression, which did not change after OT administration. These results suggest that OT can promote odontoblast-like cell differentiation, enhancing dentin formation, and that OT could be an important factor for dentinogenesis.

Family with sequence similarity 83 members H (*Fam83h*) is essential for dental enamel formation. Mutations in *Fam83h* cause human amelogenesis imperfecta (AI), an inherited disorder characterized by severe enamel hardness defects. Previous studies reported no enamel defects in *Fam83h*-KO/*lacZ*-knockin (KI) mice. Zhang et al. [228] generated genome-edited rabbits with a large deletion (900 bp) in the *Fam83h* gene using a dual single-guide RNA (sgRNA)-directed CRISPR/Cas9 system. The resulting *Fam83h* KO (*Fam83h*−/−) rabbits exhibited abnormal tooth mineralization and loose dentine. Additionally, they showed reduced hair follicle counts in dorsal skin, hair cycling dysfunction, and hair shaft differentiation. This novel *Fam83h*-KO rabbit provides a useful model for investigating *Fam83h* function and the pathogenic mechanisms of *Fam83h* mutations.

Hypodontia (dental agenesis) is a genetic disorder, and the mutation C175T in paired box 9 (*PAX9*) can cause hypodontia [229]. Liu et al. [229] investigated whether Cas9 nickase (nCas9)-mediated homology-directed repair (HDR) or the adenine base editor ABE8e could correct this *PAX9* mutation. To deliver naked DNA efficiently into DPSCs, they used a chitosan hydrogel. ABEs convert A-T base pairs to G-C base pairs without requiring double-stranded DNA breaks or donor DNA templates and thus have therapeutic potential for genetic diseases [230]. ABE8e is a recently evolved ABE variant [231]. Liu et al. [229] first established DPSC lines stably carrying a *PAX9* mutation and delivered either an HDR or ABE8e system into these cells. When gene correction efficiency was evaluated, ABE8e showed significantly higher efficiency in correcting C175T compared with HDR. Furthermore, corrected *PAX9* enhanced DPSC viability and differentiation capacity into osteogenic and neurogenic lineages. This study demonstrates that base editor systems can be effective for correcting genes related to hypodontia in DPSCs.

## 7. Effect of DPSCs on Cancer Development

MSCs interact extensively with cancer cells as key components of the tumor microenvironment [232]. DPSCs and their derivatives, such as conditioned media and cell lysates, may exert anti-cancer effects by inhibiting tumor growth and promoting apoptosis in certain cancer types [233,234]. For example, He et al. [234] demonstrated that DPSC lysate significantly inhibited the growth, migration, and invasion of A549 lung cancer cells in vitro, and this effect was also confirmed in vivo. In tumor-bearing mice, treatment with DPSC lysate suppressed tumor growth and reduced tumor weight. Furthermore, the lysate increased the expression of pro-apoptotic proteins, such as CASP3 and BCL2-associated X (BAX), while decreasing the expression of anti-apoptotic protein BCL-2, thereby triggering the intrinsic, mitochondria-mediated apoptosis pathway. Based on these results, He et al. [234] concluded that DPSC lysate may serve as an alternative to cell-based therapy for lung cancer treatment. Nikkhah et al. [233] examined whether conditioned medium (CM) derived from DPSCs (DPSC-CM) could affect the growth and migration of colorectal cancer (CRC) cells. Their experiments demonstrated that DPSC-CM significantly reduced the viability and induced the apoptosis of CRC cells through the MAP kinase and apoptosis signaling pathways. Notably, the effect of DPSC-CM was concentration-dependent. Lower concentrations potentially promoted tumor cell proliferation, whereas higher concentrations inhibited it. These findings suggest that DPSCs may play a dual role in cancer development.

## 8. Exosomes and Cytokines Secreted from DPSCs

Stem cells secrete a complex mixture of factors called the secretome, which includes exosomes and cytokines that play key roles in regenerative therapy. Exosomes are tiny vesicles that transport a cargo of proteins, lipids, and nucleic acids (such as microRNAs) between cells, thereby mediating much of the regenerative activity of stem cells [235]. Cytokines are signaling proteins, such as growth factors, chemokines, and anti-inflammatory mediators, that act as messengers between cells to regulate immune responses and tissue repair [236]. The secretions from stem cells, particularly exosomes and cytokines, are powerful modulators of wound healing and tissue repair. They can regulate immune activity, stimulate the growth of new blood vessels (angiogenesis), and promote cell survival and regeneration [237]. Based on these findings, “cell-free” therapies that utilize stem cell–derived exosomes are now considered a promising and potentially safer alternative to whole stem cells for regenerative medicine [238]. For example, Zhang et al. [239] conducted a systematic review of the role of stem cell-derived exosomes in the repair of spinal cord injury (SCI) using animal experiments. Their results indicated that stem cell-derived exosomes significantly improved motor function in animals with SCI and increased the expression of anti-inflammatory cytokines IL-4 and IL-10, as well as anti-apoptotic protein BCL-2. In contrast, they significantly lowered the levels of pro-inflammatory cytokines IL-1β and TNF-α, as well as the expression of the pro-apoptotic protein BAX. However, the mechanism of exosome-mediated SCI repair, as well as the best source and dosage, remains unknown. Therefore, future research needs to standardize animal study protocols for consistency and explore the most effective strategies for exosome-based therapy in SCI. Key priorities include establishing unified protocols for isolating and purifying exosomes, improving methods for characterizing and producing them at scale, and designing more robust animal studies to better understand the optimal delivery, dosage, and mechanisms of action.

## 9. Immunomodulatory Effects of DPSCs

To date, MSCs have been shown to exhibit immunomodulatory abilities through the expression of enzymes, the production of soluble factors, and direct cell-to-cell contacts to influence both innate and adaptive immune responses [240]. In other words, MSCs suppress various immune cells, including T cells, B cells, and macrophages, and promote a balanced, less inflammatory response [241]. These immunomodulatory abilities are also seen when human DPSCs are co-cultivated with immunogenic cells. In other words, they exhibit immunomodulatory abilities by suppressing the immune response [242]. For example, co-culturing of DPSCs and phytohemagglutinin (PHA)-activated T cells inhibited T cell proliferation and induced T cell apoptosis and the formation of regulatory T cells (Tregs), which play a central role in immune homeostasis [243,244]. DPSCs suppressed the proliferation of concanavalin A (Con A)-treated peripheral blood mononuclear cells (PBMCs), which are immune cells stimulated with Con A, in co-cultures using transwells [245]. This finding suggests that DPSCs also achieve immunomodulatory effects via the expression of soluble factors that regulate the functions of immune cells. Indeed, it was reported that DPSCs upregulated the secretion of anti-inflammatory factors, such as IL-6, TGF-β, HGF, and IL-10, in T cells, whereas they downregulated the production of pro-inflammatory factors, such as IL-2, IL-12, and TNF-α [244].

## 10. Application of Engineered DPSCs in Regenerative Medicine

Human DPSC-based therapies using cells (including gene-engineered cells) or organoid-like 3D structures have been extensively assessed in experimental animal models as basic or pre-clinical studies for regenerative medicine [14]. For cell-based therapy, the route of cell injection is important. Several routes have been reported, including intravenous administration, intra-brain injection, transplantation into the hindlimb skeletal muscle [246], and direct injection of cells (or cells encapsulated with a scaffold) into the affected sites [247]. In cases involving transplantation of larger cell aggregates, such as organoids, direct placement of these structures onto the affected sites may be appropriate. In the following sections, more detailed explanations of these approaches will be provided.

### 10.1. Cell-Based Therapy Using Intact or Engineered DPSCs

DPSCs represent a promising source for cell-based therapies owing to their easy, minimally invasive surgical access, and high proliferative capacity. Research has focused on the ability of DPSCs to treat diseases using animal models. For example, El-Kersh et al. [247] first demonstrated the therapeutic potential of intravenous (IV) and intrapancreatic (IP) transplantation of human DPSCs in a rat model of STZ-induced type 1 diabetes (T1D). IP transplantation was performed by slowly injecting the cell suspension under the pancreas capsule into 10 distinct pancreatic sites using an insulin syringe with a 29-G needle. DPSCs (10^6^ cells) labeled with the membrane-bound fluorescent marker PKH26 were injected into either the pancreas (IP) or tail vein (IV) 7 days after STZ injection, and the rats were examined about one month later. Both IV and IP transplantation of DPSCs reduced blood glucose and increased the levels of rat and human serum insulin and C-peptide. The engrafted cells survived in the STZ-injured pancreas. Islet-like clusters and scattered insulin-expressing human DPSCs, identified by anti-insulin staining and PKH26 labeling, were detected in the pancreas of diabetic rats. However, further clarification is needed regarding the possibility of an immunogenic response to the injected human cells and the transdifferentiation of human DPSCs into IPCs. El-Kersh et al. [247] concluded that DPSCs may have therapeutic potential to treat patients with long-term T1D. A similar experiment was also reported by Inada et al. [248], who generated IPCs by treating DPSCs with hydrogen sulfide in a 3D organ cultivation system and subsequently transplanted them into rat models for T1D. On the other hand, Hata et al. [246] demonstrated the usefulness of transplantation of human DPSCs into the hindlimb skeletal muscle of diabetic nude mice for the treatment of diabetic polyneuropathy. DPSC transplantation significantly reduced sensory perception thresholds, delayed nerve conduction velocity, and decreased blood flow to the sciatic nerve in diabetic mice 4 weeks post-transplantation. A subset of the transplanted DPSCs localized around the muscle bundles and expressed the human VEGF and NGF genes at the transplanted site. These results suggest that DPSC transplantation may be efficacious in treating diabetic polyneuropathy via secretion of DPSC-derived factors, such as VEGF and NGF.

Mead et al. [249] examined the therapeutic benefit of transplanting DPSCs into the vitreous body of the eye after a surgically induced optic nerve crush injury in rats. Intravitreal transplantation of DPSCs promoted neuroprotection and axon regeneration of retinal ganglion cells after optic nerve injury. Sowa et al. [250] examined the ability of HGF to provide neuroprotective effects against ischemia-induced injuries. They transplanted HGF-overexpressing DPSCs into the brains of rats following transient middle cerebral artery occlusion to determine whether the transplanted cells could attenuate brain damage associated with post-ischemia/reperfusion injury. The gene-engineered DPSCs significantly inhibited microglial activation and pro-inflammatory cytokine production, along with the suppression of neuronal degeneration. These findings suggest that the administration of HGF-overexpressing DPSCs may be useful for preventing neuronal damage in the acute phase of stroke.

### 10.2. Cell-Based Dental Stem Cell Therapy Using Intact DPSCs

Gronthos et al. [11] first demonstrated that DPSCs can regenerate tooth structures by transplanting them in conjunction with hydroxyapatite/tricalcium phosphate (HA/TCP) powder into immunocompromised mice. After 1.5 months post-implantation, the resulting 3D structures contained a dentin–pulp complex, expressing the dentin-specific protein DSPP and forming tubular structures within newly generated dentin, suggesting the presence of progenitor cells in DPSCs. To improve functional effectiveness in pulp-like tissue formation, specific growth factors and scaffold materials are required. For example, when DPSCs were embedded into 3D-printed hydroxyapatite scaffolds and transplanted into immunocompromised mice, vascularized pulp-like tissue was successfully obtained [251], highlighting the angiogenic potential of DPSCs. Alternatively, pulp-like tissues with rich vascularization can be generated in vivo without using scaffolds. In this approach, scaffold-free sheet-like aggregates composed of DPSCs are combined with a thermo-responsive hydrogel and transplanted subcutaneously into immunocompromised mice. After 1.5 months post-implantation, the resulting grafts developed pulp-like tissues with extensive blood vessel networks [238,252]. Notably, this method does not require additional growth factors or scaffolds.

### 10.3. Cell-Based Dental Stem Cell Therapy Using iPSC-Derived Dental Cells for Dentin–Pulp Complex Regeneration

iPSCs/ESCs have the potential to generate various types of differentiated cells, including neurons, adipocytes, and pancreatic β cells, as well as dental cells, such as odontoblasts and osteogenic cells. In this section, we focus on the successful production of dental cells from iPSCs/ESCs. Unlike other organs with some regenerative potential, dental tissues—particularly dental enamel—have very limited regenerative ability. To address this limitation, fabrication of dentin–pulp-like organoids using iPSC-derived cells has been explored as a strategy to regenerate damaged dental tissues. This approach is referred to as dentin–pulp complex regeneration [253]. Dentin–pulp complex regeneration requires the formation of 3D organoids, which can self-develop when seeded in an extracellular matrix (ECM)—such as Matrigel, GelMA (gelatin methacryloyl, a modified biopolymer derived from gelatin), collagen, agar gel (a natural polysaccharide polymer), or synthetic polymers, like poly(lactic-co-glycolic acid) (PLGA)—that mimics scaffolds, followed by culture in medium supplemented with growth factors. Under appropriate conditions, these organoids have been shown to contain the desired differentiated cells, similar to those in the tissue of origin. A key advantage of organoids is their ability to be serially expanded (passaged) without loss of phenotype or regenerative potential, offering a powerful tool that overcomes the limitations of cells grown on flat surfaces.

In humans, tooth development requires the interplay between epithelial and mesenchymal cells. However, studies on tooth development remain limited, as epithelial stem cells are relatively difficult to obtain and maintain. iPSCs/ESCs may be an alternative source of epithelial cells. Notably, both dental epithelial cells (DECs), derived from the ectoderm, and dental mesenchymal cells (DMCs), derived from neural crest cells, have been differentiated from iPSCs [254]. Kim et al. [255,256] established protocols for isolating DECs and DMCs through cultivation of iPSCs-derived EBs in media-specific growth factors. For iPSC-derived DECs, EBs were induced into ectodermal cells by cultivation in DMEM/F12 containing N2, BMP4, and RA. After 4 days, the medium was replaced with keratinocyte serum-free medium (K-SFM; ThermoFischer Scientific) supplemented with NOGGIN and epidermal growth factor (EGF). After another 4 days, cells were cultured in K-SFM supplemented with BMP4 and EGF. For DMC differentiation, EBs were induced into neural crest cells using DMEM/F12 supplemented with N2, B27, insulin, bFGF, and EGF. To develop bioengineered teeth using DECs and DMCs derived from human iPSCs, these differentiated cells were recombined with scaffolds, such as GelMA, collagen, and agar gel, which supported induction into osteogenic, chondro-osteogenic, and odontogenic lineages during the mineralization stage (Figure 3). These cell aggregates, encapsulated with biomaterials, were transplanted beneath the kidney capsule of mice for further maturation over 16 weeks. Kim et al. [257] also reported a novel differentiation method using a transwell membrane co-culture system to isolate DECs and DMCs from iPSCs. Additionally, DEC-like stem cells can be obtained through direct co-culture of human iPSCs with Hertwig’s epithelial root sheath/epithelial rests of the Malassez (HERS/ERM) cell line [258].

Notably, DECs can potentially regenerate dental enamel by redirecting their growth to differentiate into ameloblasts. Miao et al. [259] demonstrated that overexpression of Epiprofin (Epfn), a transcription factor specifically expressed in the dental epithelium in mouse iPSCs, led to the generation of ameloblasts, which are usually lost after tooth eruption. The presence of ameloblasts was characterized by positive staining for keratin 14 and amelogenin and Alizarin Red S staining. These results suggest that Epfn is a promising target for inducing ameloblast differentiation, which can be applied in enamel and tooth regeneration. Hemeryck et al. [260] developed a novel epithelial organoid model (enriched with cells exhibiting a tooth epithelial stemness phenotype) derived from human tooth to regenerate dental enamel. These organoids were capable of undergoing an ameloblast differentiation process, which was further enhanced by TGFβ and inhibited by TGFβ receptor blockade, thereby reproducing TGFβ’s established role in amelogenesis. Kim et al. [261] attempted to generate iPSC-derived ameloblast organoids (AOs) using a 3D culture system. The resulting AOs had similar properties to ameloblasts, forming enamel in response to calcium and undergoing mineralization through interactions with the dental mesenchyme. AOs also retained osteogenic and odontogenic differentiation potential. Furthermore, AOs demonstrated regenerative potential when combined with mouse dental mesenchyme. These findings highlight the promise of AOs not only for tooth regeneration but also for studying dental diseases that currently lack effective treatments.

Dentin–pulp complex regeneration can also be achieved in vivo. This approach focuses on injecting stem cells into disinfected root canals with an apical opening. In this case, scaffolds are required to prevent cell migration to surrounding tissues [262]. For example, dos Santos et al. [263] performed in vivo dentin–pulp complex regeneration in immunocompromised rats. After pulpectomy and canal preparation, the teeth received either DPSCs embedded in scaffolds or PBS as a control, and the cavities were sealed. After 12 weeks, the animals were euthanized, and the specimens underwent histological evaluation for intracanal connective tissue, odontoblast-like cells, intracanal mineralized tissue, and periapical inflammatory infiltrate. Transplantation of human DPSCs promoted partial pulp tissue neoformation in adult rat molars, with remnants of mineralized tissue and DMP-1-positive odontoblast-like cells.

## 11. Conclusions and Challenges

DPSCs contain multipotent stem cells with the potential for self-renewal, multilineage differentiation, and immunomodulatory functions. One of the intriguing aspects of making the use of DPSCs more attractive is their easy access from the discarded third molar tooth and minimal ethical concerns during procurement. In this context, these stem cells are more convenient than BMSCs. More importantly, DPSCs do not form tumors, and the use of autologous cells reduces the chances of rejection. For these reasons, DPSCs are considered a promising cell source for regenerative medicine applications, and several achievements have been made in their study. However, several challenges remain to be overcome. For example, DPSCs comprise a mixture of different cell types, which remains a key concern. Single-cell cloning or FACS-based cell sorting is sometimes needed, but these approaches are difficult due to the lack of standardized culture protocols, including optimal media and substrates. Production of immortalized DPSCs may be a promising approach to overcome these issues. The use of feeder cells may also help avoid the potential loss of DPSC integrity and their multilineage differentiation ability during continuous cultivation [219]. SCIP is another attractive method for obtaining a large number of cells through a one-step cultivation process of isolated primary DPSCs, as the cells show no overt alteration in their properties, even after the 10th cell passage [31].

The usefulness of cell-based therapy with human DPSCs has been highlighted by El-Kersh et al. [247], who first performed intravenous and intrapancreatic transplantation of human DPSCs in a rat model for T1D. Both approaches reduced blood glucose levels and increased the levels of rat and human serum insulin 28 days after STZ injection. Furthermore, islet-like clusters and scattered insulin-expressing human DPSCs were detected in the pancreas of diabetic rats. These findings suggest that transplanted human DPSCs may have been converted into functional IPCs, although the mechanism underlying this event remains to be elucidated. Concerns remain regarding how long the grafted cells can survive and continue secreting insulin and how strong the immunogenic response to the human cells may be since human cells were introduced into rats. In this context, more pre-clinical trials are still needed to establish the safety and efficacy of DPSC-based interventions in regenerative medicine.

DPSCs can be induced to differentiate into appropriate specific cell types under appropriate induction conditions. These include odontoblasts, osteoblasts, adipocytes, neuronal cells, hepatocytes, endothelial cells, cardiac cells, pancreatic β cells, and bladder smooth muscles, as shown in Table 2. Gene delivery can further enhance the multilineage differentiation ability of DPSCs, which is one of the promising strategies for future regenerative medicine. For example, as shown in Table 3, enhanced differentiation into odontoblasts, osteoblasts, cell proliferation, angio-/vasculogenic commitment, neurogenic commitment, adipogenic commitment, cell migration, enhanced production towards IPCs, and skeletal myogenic differentiation have all been achieved. Gene delivery has also been associated with effects on cell apoptosis and inflammation in DPSCs. Transplanting IPCs derived from the gene-engineered DPSCs into rat models of T1D has the potential to improve the diabetic phenotype. However, as with direct in vivo transplantation of DPSCs or their derivatives, challenges remain, including variability in IPC differentiation methods, the risk of immunogenic responses, and the need for long-term studies to ensure stable and effective outcomes.

iPSCs and their derivatives (i.e., iTSC and differentiated cells) generated from DPSCs are also promising resources for regenerative medicine, although their tumorigenic potential remains a concern. The main advantage of using iPSCs and their derivatives is the feasibility of obtaining these cells in large quantities. Moving toward clinical application, dentin–pulp complex regeneration is currently the most advanced application of dental stem cells. At present, both DECs and DMCs can be derived from iPSCs by inducing their differentiation with specific growth factors and chemicals [256] or using culture conditions that mimic natural biological processes, such as transwell cultivation methods [257]. The development of scaffolds is also critical for dentin–pulp complex regeneration, as they provide the structural support and environment necessary for cell adhesion, proliferation, and differentiation. Scaffolds can be derived from natural materials, such as collagen, agar, and Matrigel, or from synthetic polymers, such as PLGA.

In summary, DPSCs represent a valuable alternative to MSCs for regenerative medicine because of their easy access and minimal ethical concerns. Gene-engineered DPSCs and their derivatives (i.e., iPSCs and differentiated progeny) are also promising candidates for future cell-based therapies. Recent advances in genome editing technologies and 3D organ cultivation systems offer new opportunities for this field, including oral regeneration, by enabling more precise genetic corrections and the creation of complex, functional tissues.

## Figures and Tables

**Figure 1 biotech-14-00088-f001:**
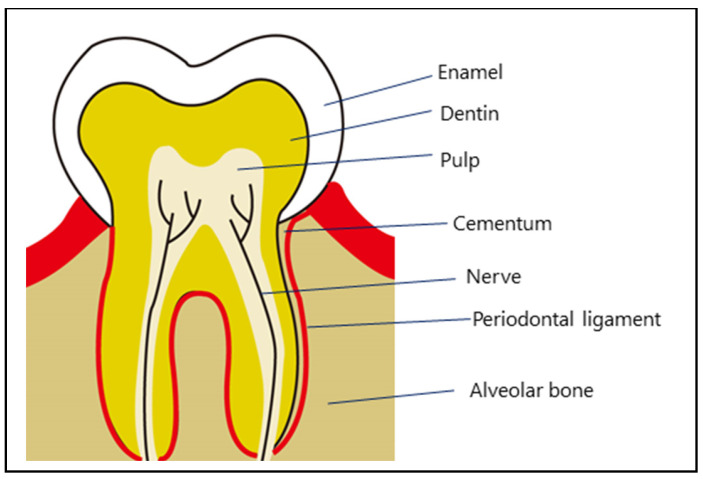
Structure of a human tooth. Dental pulp is enclosed in the dental cavity of each tooth, called the “pulp chamber,” and contains various types of cells, including fibroblasts, endothelial cells, neurons, odonto/osteo progenitors, and inflammatory and immune cells. Cells in the dental pulp are called “DPCs” and are highly enriched with stem cells called “DPSCs” with the capacity of self-renewal, rapid proliferation, multilineage differentiation, and pluripotent gene expression profile. The figures were created in-house using Adobe Illustrator version 12 (Illustrator CS2) (Adobe Inc., San Jose, CA, USA).

**Figure 2 biotech-14-00088-f002:**
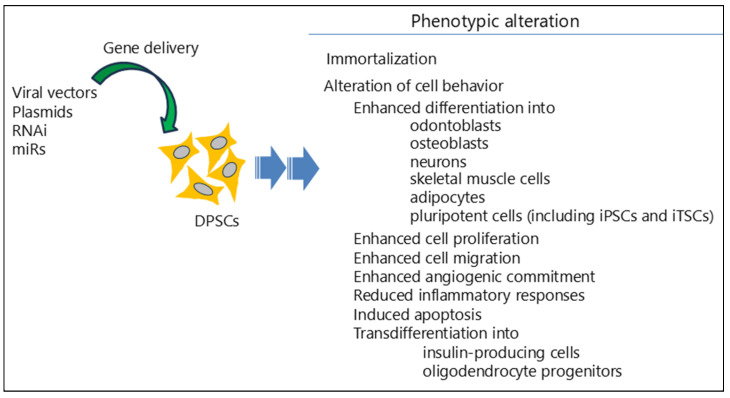
Summary of recent achievements in gene-engineered DPSCs.

**Figure 3 biotech-14-00088-f003:**
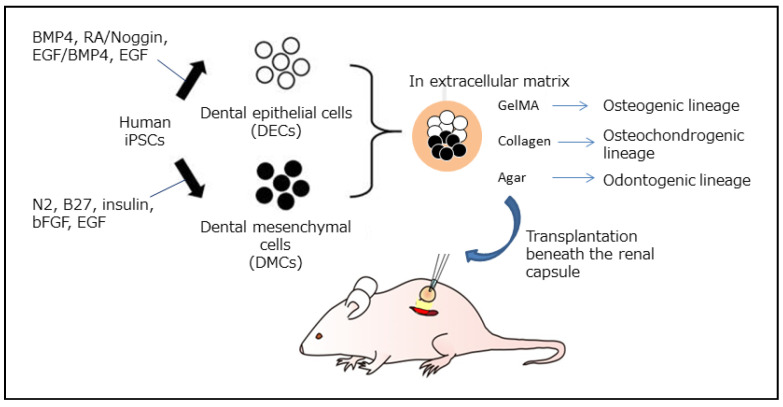
Schematic illustration of strategies for generating hard tissue from human iPSCs using extracellular matrix (ECM) microenvironments, such as gelatin methacryloyl (GelMA), agar, and collagen gel. Dental epithelial cells (DECs) and dental mesenchymal cells (DMCs) are first induced to differentiate from human iPSCs using specific humoral factors. To create bioengineered tooth-like constructs, these two types of cells are combined in the presence of various scaffolds, such as GelMA, agar, and collagen gel. The resulting mixture is subsequently embedded into a 3D-printed hydroxyapatite scaffold or deproteinized tooth scaffold, followed by transplantation beneath the renal capsule or under the skin of immunocompromised animals for further maturation. The figures were created in-house using Adobe Illustrator version 12 (Illustrator CS2) (Adobe Inc., San Jose, CA, USA) based on Kim et al. [256].

**Table 1 biotech-14-00088-t001:** Comparison of the functions between DPSCs and BMSCs.

Feature	DPSC (Dental Pulp Stem Cell)	BMSC (Bone Marrow Stem Cells)
Origin	Neural crest originates from the dental pulp tissue.	Mesodermal origin from the bone marrow.
Multilineage potential	Can differentiate into multiple cell neural lineages (osteogenic, adipogenic, chondrogenic) plus specific odontogenic and neurogenic differentiation.	Can differentiate into multiple cell neural lineages (osteogenic, adipogenic, chondrogenic), with a strong predisposition for osteogenic differentiation.
Surface markers	Typically positive for STRO-1, CD29, CD44, CD73, CD90, CD105, CD59, and CD106.	Typically positive for CD29, CD44, CD73, CD90, CD105, CD166, and CD146; lower STRO-1 expression than DPSCs.
Proliferation rate	High proliferation rate and clonogenic potential.	Slower proliferation rate compared to DPSCs.

**Table 2 biotech-14-00088-t002:** Summary of differentiation induction of human DPSCs by the methods based on non-gene-engineered approaches.

Type of Differentiated Cells, Originating from DPSCs	Methods for Differentiation Induction	Markers or Staining Method	References
Osteogenic differentiation	Cultivation in medium containing dexamethasone, L-ascorbic acid, and β-glycerol phosphate; co-cultivation with endothelium; cultivation in the presence of human serum, concentrated growth factor exudate (CGFe), and TGF-β1 or betaine; cultivation with scaffolds, decellularised adipose tissue solid foams, or microcapsules	Alkaline phosphatase (ALP) Collagen type I (COL I) Osteocalcin (OCN) Osteonectin (ON) Osteopontin (OPN) Osterix (OSX) Runt-related transcription factor 2 (RUNX2)	[47,48,49,50,51,52,53,54,55,56,57,58]
Odontogenic differentiation	Cultivation in medium containing TGF-*β*1, dexamethasone, *β*-glycerophosphate, and L-ascorbic acid; cultivating on dentin, calcium silicate materials, or scaffolds; cultivation in preameloblast-conditioned medium or medium containing endothelin-1 (ET-1) or vascular endothelial growth factor A (VEGFA)	ALP Dentin sialoprotein (DSP) Dentin sialophosphoprotein (DSPP), Dentin matrix protein 1 (DMP-1) Matrix extracellular phosphoglycoprotein (MEPE) Alizarin red S staining	[59,60,61,62,63,64,65,66]
Adipogenic differentiation	Cultivation in medium containing insulin, dexamethasone, indomethacin, and 3-isobutyl-1-methylxanthine (IBMX); cultivation in the presence of enzymatically decellularized adipose tissue solid foams	Peroxisome proliferator-activated receptor γ (PPAR-γ) Glucose transporter type 4 (GLUT4) Fatty acid binding protein 4 (FABP4) Lipoprotein lipase (LPL) Oil red O staining	[52,67,68,69,70]
Neurogenic differentiation	Cultivation in neuroinduction medium containing B27, L-glutamine, basic FGF, and EGF for 7 days, and subsequently in neuroinduction medium supplemented with retinoic acid for another 7 days; cultivation in commercially available neurogenic differentiation medium, conditioned medium of cerebrospinal fluid and retinoic acid, or medium containing nerve growth factor (NGF); culturing in the presence of graphene–polycaprolactone hybrid nanofibers or highly concentrated K^+^ (50 mM KCl) for K^+^ stimulation	Neuronal nuclei (NeuN) Microtubule-associated protein 2 (MAP2) Neural cell adhesion molecule (NCAM) Growth-associated protein 43 (GAP43) Glial fibrillary acid protein (GFAP) Synapsin I (SYN1) Neuron-specific class III beta-tubulin (TUBB3) Gamma-aminobutyric acid (GABA receptors) Enolase 2/neuron-specific enolase (ENO2/NSE) Nestin (NES) Peripherin (PRPH)	[71,72,73,74,75,76,77,78]
Hepatogenic differentiation	Cultivation in hepatoinduction medium containing hepatic growth factor, insulin-transferrin-selenium-x, dexamethasone, and oncostatin M	Alpha fetoprotein (AFP) Albumin (ALB) Hepatic nuclear factor-4 alpha (HNF4α) Insulin-like growth factor-1 (IGF-1) Carbamoyl phosphate synthetase (CPS)	[79]
Angiogenic differentiation	Cultivation in a medium supplemented with a mixture of B27, heparin, and growth factors, including vascular endothelial growth factor (VEGF)-A165 or VEGF alone; co-cultivation with human umbilical vein endothelial cells (HUVECs) after encapsulation by a scaffold system, called self-assembling peptide nanofibers	CD54/intercellular adhesion molecule-1 (ICAM-1) CD146/melanoma cell adhesion molecule (MCAM) Monocyte chemotactic protein-1 (MCP-1) von-Willebrand factor (VWF) (domains 1 and 2) CD31/platelet/endothelial cell adhesion molecule-1 (PECAM-1) Vascular endothelial growth factor (VEGF) CD34 Functioning legal knowledge 1 (Flk-1) Vascular endothelial growth factor receptor 2 (VEGFR-2) Fibroblast growth factor 2 (FGF-2) Insulin-like growth factor binding protein-3 (IGFBP3) Interleukin-8 (IL-8) Plasminogen activator inhibitor-1 (PAI-1) Tissue inhibitors of metalloproteinase-1 (TIMP-1) Urokinase-type plasminogen activator (uPA)	[80,81,82,83]
Cardiogenic differentiation	DPSCs are first incubated in the presence of 5-azacytidine for 2 days; then, the resulting embryoid bodies (EBs) are plated onto gelatin-coated tissue culture dishes, through which functional cardiomyocytes with cardiac markers are developed	Myosin heavy chain 6 (MYH6) Mesoderm posterior BHLH transcription factor 1 (MESP) NK-2 transcription factor related, locus 5 (Nkx2.5) Connexin 43 *(Cx43*)	[84]
Differentiation into pancreatic lineage	CD117^+^ DPCs have the ability to differentiate into pancreatic lineage, following the 3-step induction protocol used for pancreatic cell lineage induction; cultivation of DPSCs using a stepwise protocol to generate islet-like cell clusters; cultivation of DPSCs in a 3D culture system using a stepwise protocol to generate organoid-like 3D structures that are similar to pancreatic islets	Glucose transporter 2 (GLUT2) Pancreatic forkhead box protein A2 (Foxa2) SRY-box transcription factor 17 (Sox17) Insulin-like growth factor I (IGF-1) Fibroblast growth factor 10 (FGF 10) Pancreatic and duodenal homeobox 1 (PDX1) Hematopoietically expressed homeobox (HHEX) Motor neuron and pancreas homeobox 1 (MNX1) Neurogenin 3 (NGN3) Paired box 4 (PAX4) Paired box 6 (PAX6) NK6 homeobox 1 (NKX6-1) Blood glucose Serum insulin (INS) c-peptide (CP) Visfatin (VF) Pancreatic glucagon (GC) Somatostatin (SS) Pancreatic polypeptide, Amylase-2a (AMY2A)	[85,86,87,88,89,90,91,92]
Differentiation into bladder smooth muscle	Incubation of DPSCs in bladder smooth muscle cell-conditioned medium with transforming growth factor beta 1 (TGF-β1) for 2 weeks; incubation of DPSCs with direct contact with endothelial cells in the presence of TGF-β1	Alpha smooth muscle actin (α-SMA) Smooth muscle protein 22α (SM22α) Smooth muscle myosin heavy chain (SM-MHC) Desmin (DES) Calponin (CNN1)	[93,94,95]

**Table 3 biotech-14-00088-t003:** Summary of experiments using gene-engineered DPSCs showing gain-of-function (via overexpression) or knockdown-of-function (via RNAi or microRNA (miR)).

Biological Systems	Type of Gene Expression	Target Gene or Transgene (Method for Gene Delivery)	Cell Species	Outcome	References
Differentiation into odontoblast/mineralization	Gain-of-function (overexpression)	Growth/differentiation factor 11 (Gdf11) (electroporation)	Mouse	In vivo transfer of *Gdf11* stimulated the reparative dentin formation during pulpal wound healing in canine teeth	[105]
Differentiation into odontoblast/mineralization	Gain-of-function (overexpression)	Jagged canonical Notch ligand 1 (Jagged-1; JAG-1) (retrovirus)	Human	Overexpression of JAG-1 caused activation of the Notch signaling pathway, inhibition of odontoblastic differentiation, and the formation of mineralized tissues	[119]
Differentiation into odontoblast/mineralization	Gain-of-function (overexpression)	Krüppel-like factor 4 (KLF4) (plasmid, FuGENE)	Human	Overexpression of KLF4 caused increased alkaline phosphatase (ALP) activity and expression of odontoblastic differentiation markers	[120]
Differentiation into odontoblast/mineralization	Small interfering RNA (siRNA)-based knockdown	Notch ligand Delta1 (lentivirus)	Human	Suppression of Notch ligand Delta1 resulted in inhibition of proliferation but increased differentiation into odontoblasts	[121]
Differentiation into odontoblast/mineralization	miR-based knockdown	miR-720 (lipofectamine)	Human	miR-720, which targets *NANOG*, promoted odontogenic differentiation through suppression of *NANOG*	[122]
Differentiation into odontoblast/mineralization	siRNA-based knockdown	CD44 [HCAM (homing cell adhesion molecule)] (lentivirus)	Human	RNAi for CD44, which is expressed in odontogenic cells, resulted in suppression of mineralization activities	[123]
Differentiation into odontoblast/mineralization	Gain-of-function (overexpression)	DNA damage-inducible transcript 3 (DDIT3) (lentivirus)	Human	Overexpression of DDIT3, an apoptotic transcription factor, increased calcium nodule formation related to odontoblastic differentiation	[124]
Differentiation into odontoblast/mineralization	Gain-of-function (overexpression) siRNA-based knockdown	Inhibitor of DNA binding 1 (ID1) (lentivirus)	Human	Overexpression of ID1 resulted in an enhanced odontogenic differentiation; ID1 silencing produced an opposite effect	[125]
Differentiation into odontoblast/mineralization	Gain-of-function (overexpression) siRNA-based knockdown	Zinc finger and BTB domain-containing 20 (ZBTB20) (lentivirus)	Human	Inhibition of ZBTB20 reduced odontogenic differentiation, while overexpression of ZBTB20 enhanced odontogenic differentiation along with increased ALP activity	[126]
Differentiation into odontoblast/mineralization	Gain-of-function (overexpression)	B cell lymphoma 2 gene (BCL2) (lentivirus)	Human	Overexpression of BCL2 caused decreased osteogenic/odontogenic potential	[127]
Differentiation into odontoblast/mineralization	Gain-of-function (overexpression)	SRY-box 2 (SOX2) (retrovirus)	Human	Overexpression of SOX2 resulted in enhanced odontoblast differentiation, which was also associated with the activation of the Wnt signaling pathway	[128]
Differentiation into odontoblast/mineralization	Gain-of-function (overexpression)	Platelet-derived growth factor-BB (PDGF-BB) (lentivirus)	Human	PDGF-BB enhanced odontoblastic differentiation; subcutaneous grafting of PDGF-BB-expressing DPSCs generated dentin-like mineralized tissue	[99]
Differentiation into odontoblast/mineralization	Gain-of-function (overexpression) siRNA-based knockdown	Sclerostin (SOST) (lentivirus)	Human	Overexpression of SOST, a protein that acts as a negative regulator of bone formation, resulted in promoted senescence-related decline of odontoblastic differentiation potential; knockdown of SOST enhances odontoblastic differentiation	[129]
Differentiation into odontoblast/mineralization	siRNA-based knockdown	BTB/POZ domain-containing protein 7 (BTBD7) (lentivirus)	Human	Knockdown of *BTBD7*, a regulatory gene that promotes epithelial tissue remodeling and branching morphogenesis, resulted in reduced expression of odontoblast markers	[130]
Differentiation into odontoblast/mineralization	miR-based knockdown	miR-675 (lentivirus)	Human	Overexpression of miR-675, which is involved in odontogenic differentiation, resulted in enhancement of odontogenic differentiation	[131]
Differentiation into odontoblast/mineralization	miR-based knockdown	miR-508-5p (lipofectamine)	Human	Overexpression of miR-508-5p, which targets the glycoprotein non-metastatic melanoma protein B (*GPNMB*) gene, resulted in suppression of odontogenesis; ectopic expression of *GPNMB* (lacking 3′-UTR) rescued the effects of miR-508-5p on odontogenic differentiation	[132]
Differentiation into odontoblast/mineralization	miR-based knockdown	miR-223-3p (lentivirus)	Human	Overexpression of miR-223-3p, one of the inflammation-induced miRs, resulted in increased production of odontogenic markers but decreased the production of SMAD family member 3 (SMAD3), a potential target of miR-223-3p	[133]
Differentiation into odontoblast/mineralization	Gain-of-function (overexpression)	Dentin matrix protein 1 (DMP-1) (lentivirus)	Human	Overexpression of DMP-1 in the DPSCs derived from X-linked hypophosphatemia (XLH), associated with deficient dentin formation and mineralization, restored the irregular protein processing patterns to near-physiological levels	[134]
Differentiation into odontoblast/mineralization	Gain-of-function (overexpression) siRNA-based knockdown	Long noncoding RNA H19 (LncRNA H19) (plasmid; lentivirus)	Human	Overexpression of LncRNA H19 promoted enhanced ALP activity and increased expression of odontogenic markers via regulating the TGF-*β*1/Smad signaling pathway	[135]
Differentiation into odontoblast/mineralization	Gain-of-function (overexpression)	DMP-1 (lentivirus)	Human	Expression of DMP-1, a factor stimulating canonical Wnt signaling, in the XLH-DPSCs resulted in downregulation of Wnt inhibitors and improved mineralization	[136]
Differentiation into odontoblast/mineralization	miR-based knockdown	miR-15b-5p (lipofectamine)	Human	MiR-15b-5p suppressed the differentiation into odontoblasts by targeting insulin-like growth factor 1 (*IGF1*), while the miR-15b-5p inhibitor enhanced the differentiation into odontoblasts	[137]
Osteogenic differentiation	Gain-of-function (overexpression)	BCL2 (lentivirus)	Human	Overexpression of BCL2 resulted in enhanced cell survivability because of the inhibition of apoptosis by BCL2; it also caused decreased osteogenic/odontogenic potential	[127]
Osteogenic differentiation	Gain-of-function (overexpression)	SOX2 (retrovirus)	Human	Overexpression of SOX2 resulted in enhanced osteogenic differentiation, together with the activation of the Hippo signal pathway	[138]
Osteogenic differentiation	Gain-of-function (overexpression)	Transforming growth factor beta 1 (TGF-*β*1) (plasmid; electroporation)	Human	Overexpression of TGF-*β*1 resulted in increased osteogenic and chondrogenic differentiation; it also resulted in increased proliferation rate and decreased apoptosis	[111]
Osteogenic differentiation	Gain-of-function (overexpression)	Wnt family member 4 (WNT4) (lentivirus)	Human	Overexpression of WNT4 exhibited effective repair of rat bone defects, suggesting that WNT4-expressing DPSCs may be a feasible resource of seed cells for bone regeneration	[139]
Osteogenic differentiation	Gain-of-function (overexpression)	EphrinB2 (EFNB2) (lentivirus)	Human canine	Overexpression of EFNB2 resulted in enhanced osteogenic differentiation capacity	[101]
Osteogenic differentiation	Gain-of-function (overexpression) miR-based knockdown	hsa_circ_0026827 miR-188-3p (plasmid; lipofectamine)	Human	Overexpression of hsa_circ_0026827, one of the circular RNAs (circRNAs), resulted in increased osteoblast differentiation, while knockdown of hsa_circ_0026827 suppressed osteoblast differentiation and promoted miR-188-3p expression	[140]
Osteogenic differentiation	Gain-of-function (overexpression)	Hypoxia-inducible factor 1α (HIF-1α) (protein transduction domains (PTDs)) (lentivirus)	Human	Overexpression of HIF-1α, a protein known to be expressed in hypoxia and affect stemness and bone differentiation, resulted in enhanced osteogenic differentiation	[141]
Osteogenic differentiation	Gain-of-function (overexpression)	Semaphorin 3A (SEMA3A) (lentivirus)	Unknown	Overexpression of Sema3A, a secretory osteoprotective factor, exhibited enhanced the osteogenic differentiation	[142]
Osteogenic differentiation	Gain-of-function (overexpression) siRNA-based knockdown	Growth differentiation factor 15 (GDF15) (lentivirus; lipofectamine)	Human	Overexpression of GDF15 caused enhanced osteogenic differentiation through activation of the TGF-β/SMAD signaling pathway, while knockdown of GDF15 produced the opposite effect	[143]
Enhanced cell proliferation	siRNA-based knockdown	Catalytic alpha1 of AMP-activated protein kinase (AMPK alpha1) (siRNA; lipofectamine)	Rat	Knockdown of AMPKalpha1, a stress-responsive enzyme that is activated by hypoxia, resulted in decreased cell proliferation under both normoxia and hypoxia	[144]
Enhanced cell proliferation	siRNA-based knockdown	Notch ligand Delta1 (lentivirus)	Human	Knockdown of Notch ligand Delta1 resulted in inhibition of proliferation but increased differentiation into odontoblasts	[121]
Enhanced cell proliferation	miRNA-based knockdown	miR-720 (lipofectamine)	Human	miR-720, targeting *NANOG*, caused decreased proliferation and promoted odontogenic differentiation through suppression of *NANOG*	[122]
Enhanced cell proliferation	Gain-of-function (overexpression)	DDIT3 (lentivirus)	Human	Overexpression of DDIT3, an apoptotic transcription factor, resulted in reduced cell proliferation but increased calcium nodule formation related to odontoblastic differentiation	[124]
Enhanced cell proliferation	Gain-of-function (overexpression)	SOX2 (retrovirus)	Human	SOX2 overexpression resulted in the enhancement of cell proliferation, migration, and adhesion	[145]
Enhanced cell proliferation	miR-based knockdown	MiR-633 (lentivirus)	Human	miR-633 overexpression increased cell proliferation and differentiation through direct interaction with the 3′-UTR of matrix extracellular phosphoglycoprotein (*MEPE*) gene	[146]
Enhanced cell proliferation	Gain-of-function (overexpression)	Special AT-rich binding protein 2 (SATB2) (lentivirus)	Human	Overexpression of SATB2 resulted in accelerated proliferation and cell migration	[147]
Enhanced cell proliferation	Gain-of-function (overexpression)	PDGF-BB (lentivirus)	Human	PDGF-BB caused enhanced proliferation, odontoblastic differentiation, and cell migration via the activation of the phosphatidylinositol 3 kinase (PI3K)/Akt signaling pathway	[99]
Enhanced cell proliferation	Gain-of-function (overexpression)	Vascular endothelial growth factor (VEGF)/ stromal cell-derived factor-1α (SDF-1α) (lentivirus)	Human	Overexpression of SDF-1α or VEGF resulted in enhanced cell proliferation, endothelial cell migration, and vascular tube formation on Matrigel in vitro; expression of both VEGF and SDF-1α resulted in enhanced vascularized dental pulp regeneration in vivo	[148]
Enhanced cell proliferation	Gain-of-function (overexpression)	Lin28 (lentivirus)	Human	Overexpression of Lin28, a conserved RNA-binding protein in eukaryotes, caused increased proliferation through interaction with let-7a/IGF2BP2 pathways	[149]
Enhanced cell proliferation	miR-based knockdown	miR-210-3p (lentivirus)	Human	Overexpression of miR-633, targeting the 3′ UTR of the *MEPE* gene, caused increased cell proliferation and differentiation	[150]
Enhanced cell proliferation	Gain-of-function (overexpression)	TGF-*β*1 (plasmid; electroporation)	Human	Overexpression of TGF-*β*1 resulted in increased osteogenic and chondrogenic differentiation but decreased adipogenic differentiation; it also resulted in increased proliferation rate and decreased apoptosis	[111]
Enhanced cell proliferation	siRNA-based knockdown	Visfatin (VIS) (unknown)	Human	Knockdown of VIS, a novel adipokine associated with cellular senescence, resulted in reduced senescence	[151]
Enhanced cell proliferation	Gain-of-function (overexpression)	Hepatocyte growth factor (HGF) (adenovirus)	Human	Overexpression of HGF inhibited rheumatoid arthritis progression by its immunosuppressive effects, while in the late phase, HGF promoted synovitis by activating fibroblast-like synoviocytes and exhibited accelerated cell proliferation and apoptosis resistance, suggesting a dual role of HGF in RA	[152]
Enhanced cell proliferation	Gain-of-function (overexpression)	Long noncoding RNA H19 (LncRNA H19) (plasmid; lentivirus)	Human	Overexpression of LncRNA H19 caused increased cell proliferation, enhanced ALP activity, and increased odontoblast markers via regulating the TGF-*β*1/Smad signaling pathway	[135]
Enhanced cell proliferation	Gain-of-function (overexpression)	LncRNA H19 (plasmid; lentivirus)	Human	Overexpression of LncRNA H19 promoted cell proliferation, enhanced ALP activity, and increased odontoblast markers via regulating the TGF-*β*1/Smad signaling pathway	[153]
Enhanced cell proliferation	Gain-of-function (overexpression)	Adrenomedullin (ADM) (lentivirus)	Human	Overexpression of ADM resulted in promotion of cell cycle progression, inhibition of p53 expression, decreased reactive oxygen species (ROS) accumulation, and resistance against cellular senescence	[154]
Angio-/vasculogenic commitment	miR-based knockdown	MiR-424 (lentivirus)	Human	Overexpression of miR-424 caused decreased levels of VEGF and kinase insert domain-containing receptor (KDR) protein	[155]
Angio-/vasculogenic commitment	Gain-of-function (overexpression)	PDGF-BB (lentivirus)	Human	PDGF-BB caused enhanced proliferation, odontoblastic differentiation, and cell migration via the activation of the PI3K/Akt signaling pathway	[99]
Angio-/vasculogenic commitment	Gain-of-function (overexpression)	VEGF SDF-1α (lentivirus)	Human	Overexpression of SDF-1α or VEGF resulted in enhanced cell proliferation, endothelial cell migration, and vascular tube formation on Matrigel in vitro; expression of both VEGF and SDF-1α resulted in enhanced vascularized dental pulp regeneration in vivo	[148]
Angio-/vasculogenic commitment	Gain-of-function (overexpression)	BCL-2/ green fluorescent protein (GFP) (lentivirus)	Human	Overexpression of BCL-2 resulted in enhanced endothelial cell proliferation, migration, and vascular tube formation on Matrigel, suggesting that BCL-2 overexpression enhances angio-/vasculogenic properties of DPSCs	[156]
Angio-/vasculogenic commitment	Gain-of-function (overexpression)	Ets variant transcription factor 2 (ETV2) (lentivirus)	Human	Overexpression of ETV2 resulted in the appearance of endothelial-like morphology and increased expression of endothelial-specific genes, which also correlated with enhanced capillary-like tube formation on Matrigel in vitro	[157]
Angio-/vasculogenic commitment	Gain-of-function (overexpression)	PDGF-BB (lentivirus)	Human	Co-cultivation of PDGF-BB-overexpressing cells with human umbilical vein endothelial cells (HUVECs) exhibited increased formation of vascular tubes, suggesting that PDGF-BB engineering is an effective strategy to amplify DPSCs’ angiogenic potential	[158]
Neurogenic commitment	Gain-of-function (overexpression)	zinc finger protein 521 (Zfp521) (unknown)	unknown	Overexpression of Zfp521, a transcription factor, can facilitate differentiation into neural cells through chromatin modification	[159]
Neurogenic commitment	Gain-of-function (overexpression)	Insulin-like growth factor-binding protein 5 (IGFBP5) (retrovirus)	Human	Overexpression of IGFBP5 prompted neurogenic differentiation potential of DPSCs, suggesting that IGFBP5 is a potential target for dental pulp–dentin functional regeneration	[160]
Neurogenic commitment	Gain-of-function (overexpression)	OCT3/4 (lentivirus)	Human	Overexpression of OCT3/4 under neural inductive conditions caused reprogramming into the neural lineage	[161]
Adipogenic commitment	Gain-of-function (overexpression)	Ten-eleven-translocation 2 (TET2) (plasmid; lipofectamine)	Human	Overexpression of *TET2*, a key regulator of DNA methylation during adipogenic induction, resulted in increased expression of adipogenic marker genes and enhancement of the transition of DPSCs toward adipogenic commitment	[162]
Inflammation commitment	siRNA-mediated knockdown	Receptor for advanced glycation end products (RAGE) (lipofectamine)	Human	Binding of RAGE to high-mobility group box 1 (HMGB1), a nonhistone DNA-binding protein that promotes inflammation, is directly related to eliciting inflammation	[163]
Inflammation commitment	miR-based knockdown	miR-223-3p (lentivirus)	Human	Overexpression of miR-223-3p, one of the inflammation-induced miRNAs, resulted in increased production of odontoblast marker genes; knockdown of Smad3 increased the level of ALP, thereby promoting odontoblast differentiation	[133]
Inflammation commitment	Gain-of-function (overexpression)	HGF (adenovirus)	Human	Overexpression of HGF resulted in enhanced downregulation of inflammation-related factors	[164]
Inflammation commitment	Gain-of-function (overexpression)	WNT4 (plasmid; lipofectamine)	Human	Overexpression of WNT4 resulted in amelioration of cell inflammatory response, enhanced BCL-2 expression, and decreased apoptosis rate in the DPCs inflamed by lipopolysaccharide (LPS)	[165]
Cell migration	siRNA-based knockdown	Peptidylprolyl cis/trans isomerase, NIMA-interacting 1 (PIN1) (plasmid; lipofectamine)	Human	siRNA-based silencing of *PIN1* (which specifically binds to phosphorylated Ser/Thr-pro motifs to catalytically regulate the post-phosphorylation conformation of its substrates) decreased the cell migration of DPSC	[166]
Cell migration	Gain-of-function (overexpression)	PDGF-BB (lentivirus)	Human	PDGF-BB enhanced odontoblastic differentiation and cell migration	[99]
Enhanced induction into insulin-producing cells (IPCs)	Gain-of-function (overexpression)	Forkhead box A2 (FOXA2) Pancreatic and duodenal homeobox 1 (PDX1) (lentivirus)	Human	DPSCs can be reprogrammed to generate insulin-producing cells (IPCs) by transducing them with the transcription factor genes *FOXA2* and *PDX1*	[167]
Enhanced induction into IPCs	miR-based knockdown using miR inhibitor	miR-183 inhibitor (FuGENE)	Rat	Downregulation of *miR-183* through transfection with miR-183 inhibitor resulted in the generation of cells expressing insulin 72 h after transfection	[168]
Enhanced induction into IPCs	Gain-of-function (overexpression)	Pdx1 Neurogenin 3 (Neurog3) (plasmid; FuGENE)	Rat	Transfection with vectors carrying transcription factors *Pdx1* and *Neurog3* caused direct conversion into IPCs	[169]
Enhanced induction into oligodendrocyte progenitors	Gain-of-function (overexpression)	Oligodendrocyte transcription factor 2 (Olig2) (plasmid; X-tremeGENE)	Human	Transfection of DPSCs with the *Oligo2* gene and subsequent cultivation in the differentiation-inducing medium resulted in the generation of cells showing oligodendrogenic markers	[170]
Cell apoptosis	RNAi (siRNA oligo)- mediated knockdown	Caspase-8 and caspase-9 (unknown)	Human	Knockdown of caspase-9 exhibited a reduction in apoptosis, caspase-3 expression, and its activity	[171]
Cell apoptosis	miR-based knockdown	miR-224-5p (lipofectamine)	Human	Inhibition of miR-224-5p, which targets the 3′-untranslated region (3′-UTR) of the Rac family small GTPase 1 (*Rac1*) gene, caused increased cell apoptosis	[172]
Skeletal myogenic differentiation	miR-based knockdown	miR-143 and miR-135 inhibitors (lipofectamine)	Human	miR-135 and miR-143 inhibitors induce myogenic differentiation of DPSCs	[173]
Skeletal myogenic differentiation	miR-based knockdown	miRNA-139-5p (lipofectamine)	Human	Overexpression of miR-139-5p induced skeletal myogenic differentiation via the Wnt/β-catenin signaling pathway; downregulation of miR-139-5p inhibited cell growth and reduced skeletal myogenic differentiation	[174]
Pluripotency and multilineage differentiation capability	Gain-of-function (overexpression)	Octamer-binding transcription factor 4A (OCT4A) (lentivirus)	Human	Overexpression of OCT4A resulted in upregulation of expression of stemness factors, as well as increased cell proliferation, pluripotency, and multilineage differentiation potential	[175]
Pluripotency and multilineage differentiation capability	Gain-of-function (overexpression)	Xeroderma pigmentosum complementation group C protein (XPC) (lentivirus)	Human	XPC, a component of the DNA repair pathway, interacts with OCT3/4 to modulate pluripotency; overexpression of XPC enhances proliferation rate, reduces apoptosis, and improves DPSC’s multilineage differentiation capabilities	[176]

## Data Availability

No new data were created or analyzed in this study.

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
