# Peer review of "Engineered Human Dental Pulp Stem Cells with Promising Potential for Regenerative Medicine"

_biotech, 2025, doi:10.3390/biotech14040088_

Round 1

Reviewer 1 Report

Comments and Suggestions for Authors

My comments are attached in the file, please check

Author Response

reviewer-1:

The review article entitled “Engineered human dental pulp stem cells with promising potential for regenerative medicine” is well written and explained the broad scope of dental stem cells for tissue engineering applications.

Question-1: It is recommended to include a table showing the surface marker expression and multilineage differentiation potential of dental pulp stem cells compared to other adult tissue-specific stem cells (e.g., bone marrow-derived stem cells, adipose tissue-derived stem cells, and other sources).

Answer-1: As suggested, a table showing comparison on the property for both DPSC (dental pulp stem cell) and BMSC (bone marrow stem cell) is created as Table 1 as new Table. 

Question-2: Why are dental pulp stem cells superior to other types of stem cells and what makes them a unique choice?

Answer-2: This point is mentioned in the revised text (please see P3 middle low).

Question-3: How do these dental pulp stem cells survive under ischemic conditions?

Answer-3: The reason why DPSCs can survive under the ischemic conditions is mentioned in the revised text (please see P15 lower tier).

Question-4: Does dental pulp stem cells any role in play in cancer development?

Answer-4: Yes. Some factors secreted from DPSCs or their lysate itself play a role in preventing cancer development. This point is mentioned as a new section “7. Effect of DPSC on cancer development” in the revised text (P23).   

Question-5: What is the secretory profile of these stem cells such as exosomes and cytokines, and do they have any role in regenerative therapy?

Answer-5: Exosomes and cytokines secreted from DPSCs are considered to play a role in the events related to regenerative medicine. This point is mentioned as a new section “8. Exosomes and cytokines secreted from DPSC” in the revised text (P23).

Question-6: Please include a section on the immunomodulatory effects of stem cells Based on my thorough review, I recommend minor revisions to the manuscript.

Answer-6: As suggested, a section on the immunomodulatory effects of DPSC is newly made. Please see a new section “9. Immunomodulatory effects of DPSCs” in the revised text (P24).

Reviewer 2 Report

Comments and Suggestions for Authors

The authors conducted a thorough review of the literature regarding human dental pulp stem cells that merits publication.
As the section "3. Gene Engineering of DPSCs" is large and detailed, it is recommended to create an image to summarize key finding that with help reading.
Some minor revisions are also proposed:
-"To date, various types of DPSCs have been reported, including those from stem cells 109 from human..." Rephrase this sentence. You might mean that DPSCs might be transdifferentiated by other stem cell populations.
-Figure 1, explain in caption how this image was created, tool used etc.
-Table 1, first column correct "Ondontogenic" to "Odontogenic"
-"2-6 Markers for DPSCs" format of this heading should be corrected
-Figure in page 24 covers figure caption.

Author Response

reviewer-2: Comments and Suggestions for Authors

The authors conducted a thorough review of the literature regarding human dental pulp stem cells that merits publication.

Question-1: As the section "3. Gene Engineering of DPSCs" is large and detailed, it is recommended to create an image to summarize key finding that with help reading.

Answer-1: As suggested, Figure 2 showing a whole image to summarize the content mentioned in "3. Gene Engineering of DPSCs" is newly set in the revised text.

Some minor revisions are also proposed:

Question-2: -"To date, various types of DPSCs have been reported, including those from stem cells from human..." Rephrase this sentence. You might mean that DPSCs might be transdifferentiated by other stem cell populations.

Answer-2: Thank you for your suggestion. This portion was corrected (please see P3 lower tier) in the revised text).

Question-3: -Figure 1, explain in caption how this image was created, tool used etc.

Answer-3: As suggested, this was done (please see the caption of Figures 1 and 3 in the revised text).

Question-4: -Table 1, first column correct "Ondontogenic" to "Odontogenic"

Answer-4: Sorry. This was a careless mistake. This portion is corrected. 

Question-5: -"2-6 Markers for DPSCs" format of this heading should be corrected

Answer-5: This was correct (please see L6 lower tier).

Question-6: -Figure in page 24 covers figure caption.

Answer-6: This was correct (please see P27 in the revised text).

Reviewer 3 Report

Comments and Suggestions for Authors

This review provides a comprehensive examination of dental pulp stem cells (DPSCs), thoroughly covering their isolation, culture, multilineage differentiation, genetic engineering, iPSC induction, genome editing, and clinical applications. The manuscript is supported by substantial data, with particularly valuable summaries of both non-engineered and gene-engineered differentiation approaches in Tables 1 and 2. Furthermore, it demonstrates strong relevance to current research trends through its inclusion of cutting-edge technologies such as CRISPR/Cas9, iPSCs, 3D culture, and organoid systems. However, despite these strengths, the manuscript would benefit from a clearer structural focus and a more critical synthesis of the reviewed studies to enhance its academic contribution.

For page 4, Section "2-2 Optimizing conditions for DPSCs in cell culture":

The statement that "the proliferation and differentiation of DPSCs vary considerably depending on the serum batches" is immediately followed by the assertion that human serum (HS) is more suitable for DPSC culture. This statement could be reconsidered, because fetal bovine serum (FBS) remains a more widely used and standardized reagent in cell culture, typically demonstrating better batch-to-batch consistency compared to HS. It is recommended to reorganize this section, potentially integrating it with Section "2-3 Requirement for xeno-free cell culture system," to provide a more balanced and logically coherent discussion.

For page 6-10, Section "2-7 Multi-lineage differentiation potential of DPSCs":

The scientific value could be enhanced by incorporating comparisons between DPSCs and other stem cell sources (e.g., BMSCs, ESCs) across all differentiation subsections. For instance, the osteogenic differentiation part should explicitly contrast the efficiency, marker expression, and practical advantages of DPSCs versus these other stem cells to provide critical context for their regenerative application.

For page 11–13, Table 2:

 This section lacks a synthesized analysis of the consistency or contradictions across studies. It is recommended to add a dedicated subsection (e.g., "3-4 Critical Evaluation of Gene-Engineering Strategies") to summarize the roles of key signaling pathways (e.g., Wnt, TGF-β, Notch) in regulating DPSC behavior. 

The the comprehensiveness of Table 2 is commendable,however, its density may reduce readability, I suggest retaining only the most critical findings and relocating the remainder to an appendix.

For page 20, Section "5. Usefulness of Feeder Cells to Maintain the Integrity of DPSCs":

While feeder systems represent a conventional method for stem cell culture, numerous alternative approaches (e.g., Matrigel) have been developed to enhance experimental reproducibility and eliminate variability associated with feeder cell conditions. This section should address whether such feeder-free culture methods have been successfully applied to DPSCs. If applicable, a comparison of the advantages and limitations between feeder-free and feeder-dependent systems for DPSC culture should be provided. If no effective feeder-free methods exist for DPSCs, this should be explicitly stated to substantiate the necessity of feeder layers for maintaining DPSC integrity.

Spell check

In table 2, the row citing reference [148]: The term "accerelated" should be corrected to "accelerated".

In the title of Table 1: The term: "diffrerentation" should be corrected to "differentiation".

A comprehensive spell check of the entire manuscript is recommended to eliminate remaining typographical errors.

Author Response

reviewer-3: Comments and Suggestions for Authors

This review provides a comprehensive examination of dental pulp stem cells (DPSCs), thoroughly covering their isolation, culture, multilineage differentiation, genetic engineering, iPSC induction, genome editing, and clinical applications. The manuscript is supported by substantial data, with particularly valuable summaries of both non-engineered and gene-engineered differentiation approaches in Tables 1 and 2. Furthermore, it demonstrates strong relevance to current research trends through its inclusion of cutting-edge technologies such as CRISPR/Cas9, iPSCs, 3D culture, and organoid systems. However, despite these strengths, the manuscript would benefit from a clearer structural focus and a more critical synthesis of the reviewed studies to enhance its academic contribution.

Question-1: For page 4, Section "2-2 Optimizing conditions for DPSCs in cell culture":

The statement that "the proliferation and differentiation of DPSCs vary considerably depending on the serum batches" is immediately followed by the assertion that human serum (HS) is more suitable for DPSC culture. This statement could be reconsidered, because fetal bovine serum (FBS) remains a more widely used and standardized reagent in cell culture, typically demonstrating better batch-to-batch consistency compared to HS. It is recommended to reorganize this section, potentially integrating it with Section "2-3 Requirement for xeno-free cell culture system," to provide a more balanced and logically coherent discussion.

Answer-1: For use of human DPSCs in regenerative medicine, human serum (HS) is more suitable than animal-derived serum such a fetal bovine serum (FBS). However, there are still differences in batch-to-batch serum, which are a persistent and common issue in cell culture. This point is mentioned in the revised text (please see P5 Upper tier).

Question-2: For page 6-10, Section "2-7 Multi-lineage differentiation potential of DPSCs":

The scientific value could be enhanced by incorporating comparisons between DPSCs and other stem cell sources (e.g., BMSCs, ESCs) across all differentiation subsections. For instance, the osteogenic differentiation part should explicitly contrast the efficiency, marker expression, and practical advantages of DPSCs versus these other stem cells to provide critical context for their regenerative application.

Answer-2: Thank you for your nice suggestion. In the revised text, comparison of the property between DPSCs and BMSCs is newly made as Table 1. Several points (showing that DPSCs are a superior stem cell source for obtaining neural progenitor cells, and they display stronger odontogenic capability than BMSCs) make DPSCs advantageous for tooth regeneration compared to other sources. These are mentioned in the revised text (please see P3 middle low).

Question-3: For page 11–13, Table 2:

 This section lacks a synthesized analysis of the consistency or contradictions across studies. It is recommended to add a dedicated subsection (e.g., "3-4 Critical Evaluation of Gene-Engineering Strategies") to summarize the roles of key signaling pathways (e.g., Wnt, TGF-β, Notch) in regulating DPSC behavior. The the comprehensiveness of Table 2 is commendable, however, its density may reduce readability, I suggest retaining only the most critical findings and relocating the remainder to an appendix.

Answer-3: As suggested, the section showing the synthesized analysis of DPSCs is created as "3-4 Critical Evaluation of Gene-Engineering Strategies" in the revised text. In this, the roles of key signaling pathways (e.g., Wnt, TGF-β, Notch) in regulating DPSC behavior are summarized briefly. Also, in Table 2 (now shown as Table 3 in the revised text), the content described in “Note” is shortened as possible. 

Question-4: For page 20, Section "5. Usefulness of Feeder Cells to Maintain the Integrity of DPSCs":

While feeder systems represent a conventional method for stem cell culture, numerous alternative approaches (e.g., Matrigel) have been developed to enhance experimental reproducibility and eliminate variability associated with feeder cell conditions. This section should address whether such feeder-free culture methods have been successfully applied to DPSCs. If applicable, a comparison of the advantages and limitations between feeder-free and feeder-dependent systems for DPSC culture should be provided. If no effective feeder-free methods exist for DPSCs, this should be explicitly stated to substantiate the necessity of feeder layers for maintaining DPSC integrity.

Answer-4: As suggested, there are some ways (e.g., using Matrigel) to cultivate stem cells. These are mentioned in the revised text (please see P21 middle low). Also, the advantages and limitations between feeder-free and feeder-dependent systems for DPSC culture are mentioned (please see P21 lower tier).

Question-5: Spell check

In table 2, the row citing reference [148]: The term "accerelated" should be corrected to "accelerated".

In the title of Table 1: The term: "diffrerentation" should be corrected to "differentiation".

A comprehensive spell check of the entire manuscript is recommended to eliminate remaining typographical errors.

Answer-5: The points indicated are corrected. The other part was also checked.

Round 2

Reviewer 1 Report

Comments and Suggestions for Authors

Thanks for the edits. The authors have done an excellent job and improved the manuscript.